# A 30-meter terrace mapping in China using Landsat 8 imagery and digital elevation model based on the Google Earth Engine

Bowen Cao[1], Le Yu[1,2,*], Victoria Naipal[3], Philippe Ciais[4], Wei Li[1,2], Yuanyuan Zhao[5,6], Wei Wei[7], Die Chen[7], Zhuang Liu[1], Peng Gong[1,2]

[1]Ministry of Education Key Laboratory for Earth System Modeling, Department of Earth System Science, Tsinghua University, Beijing 100084, China;
[2]Ministry of Education Ecological Field Station for East Asian Migratory Birds, Beijing 100084, China;
[3]Department of Geography, Ludwig-Maximilian University, Munich, Germany;
[4]Laboratoire des Sciences du Climat et de l'Environnement, CEA-CNRS-UVSQ, UMR8212, Gif-sur-Yvette, France;
[5]College of Land Science and Technology, China Agricultural University, Beijing 100083, China;
[6]Key Laboratory of Remote Sensing for Agri-Hazards, Ministry of Agriculture and Rural Affairs, Beijing 100083, China;
[7]State Key Laboratory of Urban and Regional Ecology, Research Center for Eco-environmental Sciences, Chinese Academy of Sciences, Beijing 100085, China

*Correspondence to*: Le Yu (leyu@tsinghua.edu.cn)

**Abstract.** The construction of terraces is a key soil conservation practice on agricultural land in China, providing multiple valuable ecosystem services. Accurate spatial information on terraces is needed for both management and research. In this study, the first 30 m resolution terracing map of the entire territory of China is produced by a supervised pixel-based classification using multi-source and multi-temporal data based on the Google Earth Engine (GEE) platform. We extracted time-series spectral features and topographic features from Landsat 8 images and the Shuttle Radar Topography Mission digital elevation model (SRTM DEM) data, classifying cropland area (cultivated land of Globeland30) into terraced and non-terraced type through a random forest classifier. The overall accuracy and kappa coefficient were evaluated by 10875 test samples and achieved values of 94 % and 0.72, respectively. For terrace class, the producer's accuracy (PA) was 79.945% and the user's accuracy (UA) was 71.149%. The classification performed best in the Loess Plateau and southwestern China, where terraces are most numerous. Some northeastern, central eastern and southern area had relatively high uncertainty. Typical errors in the mapping results from the sloping cropland (non-terrace cropland with a slope of $\geq 5°$), low-slope terraces, and non-crop vegetation. Terraces are widely distributed in China and the total terraced area was estimated to be 53.55Mha (i.e., 26.43% of China's cropland area) by pixel counting (PC) method and 58.46±2.99 Mha (i.e., 28.85%±1.48% of China's cropland area) by error matrix-based model-assisted estimation (EM) method. Elevation and slope were identified as the main features in the terrace/non-terrace classification, and multi-temporal spectral features (such as percentiles of NDVI, TIRS2, BSI) were also essential. Terraces are more challenging to identify than other land use types because of the intra-class feature heterogeneity, inter-class feature similarity and fragmented patches, which should be the focus of future research. Our terrace mapping algorithm can be used to map large-scale terraces in other regions globally, and our terrace map will serve as a landmark for studies on multiple ecosystem services assessments including erosion control, carbon

sequestration, and biodiversity conservation. The China terrace map is available to the public at
https://doi.org/10.5281/zenodo.3895585 (Cao et al., 2020).

## 1 Introduction

Building agricultural terraces is a widespread adaptive strategy for sustaining cropland agriculture in areas where water erosion, severe drought, mass movement and landslides threaten crop production, soil conservation, and man-made infrastructure (Lasanta et al., 2001). Multiple ecological functions are performed by terraces, including reducing water runoff
(Chow et al., 1999; Montgomery, 2007), controlling soil erosion (Sharda et al., 2002), improving land productivity (Homburg and Sandor, 2011), ensuring food security (Liu et al., 2011; Rockström and Falkenmark, 2015), and enhancing biodiversity as well as ecosystem restoration (Tokuoka and Hashigoe, 2014; Wei et al., 2012). Terracing may reduce soil erosion rates up to 95 % (Fu, 1989). In this way, not only soil moisture but also soil organic carbon and nutrients can be preserved. A meta-analysis for the ecosystem benefits of terracing shows that, compared to un-terraced slopes, soil on
terraced slopes contains up to 28.1% more total nitrogen and 41.7% more soil organic matter, respectively (Wei et al., 2016). Another recent meta-analysis study on terracing and soil organic carbon sequestration revealed that terracing increased SOC sequestration by 32.4 % for China (Chen et al., 2020).

Terraces are mainly found in mountainous and hilly regions (Wei et al., 2016). Since approximately two-thirds of China is covered by mountains, terrace farming is one of the predominant forms of cropland agriculture in China (Lü et al., 2009). In
recent years, China has made water and soil loss control in sloping cropland (non-terrace cropland with a slope of ≥5°) a focus of land consolidation (NDRC and MWR, 2017). One of the most common engineering measures applied in comprehensive soil conservation and improved farming projects is to transform slopes into terraces.

However, the terrace number and area distribution in China are poorly documented. Current terrace statistics are incomplete or outdated and no national map exists. Accurate and detailed spatial distribution information of terraces is thus lacking.
Therefore, it is essential to generate national-scale data to provide information for policy-makers to make decisions and take actions in land planning and agricultural management, and for researchers to evaluate the value and impact of terraces on ecosystem.

Remote sensing is often used to rapidly obtain detailed land cover and land use information (Gong et al., 2013; Yu et al., 2013b), agricultural monitoring being one critical application. In recent decades, significant progress has been made in
remote sensing-based identification of cropland areas (Pittman et al., 2010; Yu et al., 2013a), specific crop types (Bargiel, 2017; Zhong et al., 2014), farming practices such as rainfed/irrigated cropland (Jin et al., 2016), multiple cropping (Biradar and Xiao, 2011) and fallow (Estel et al., 2015). A few studies have insofar explored the possibility of using remote sensing to map terraces. Visual interpretation based on satellite images is the conventional method (Faulkner et al., 2003; Martínez-Casasnovas et al., 2010), mainly based on visual characteristics and empirical knowledge to map the terraces. A few
automated approaches involving object-based classification have been proposed (Diaz-Varela et al., 2014; Zhao et al., 2017),

which recognize terraces based on features of individual objects. Other studies applied texture analysis methods such as Fourier transformation (Zhang et al., 2017) and gray-level co-occurrence matrix analysis (Li et al., 2013) to identify terraces. While the results seem to be promising, such studies were limited to small areas. These algorithms were low efficiency and required case-specific parameters, which cannot be easily scaled to large areas.

In this study, we establish the first classification of agricultural terraces over China for the epoch of the year 2018. Our method is a supervised pixel-based classification, which has been widely used for other applications, showing good performance for large-scale cropland mapping (Gumma et al., 2020; Teluguntla et al., 2018). Thus, it has potential in large-scale terrace/non-terrace classification. However, few studies have explored its application in the field. Here, we adapted and extended this approach to systematic agricultural terrace mapping.

As for the selected data for classification of terrace/non-terrace, usually, past studies relied on images from high-resolution satellites such as WorldView (Luo et al., 2020; Zhao et al., 2017), SPOT-5 (Li et al., 2013), Gaofen-1 (Zhang et al., 2017) and unmanned aerial vehicles (Diaz-Varela et al., 2014). Such data are expensive and are not universally accessible, making them unsuitable for large-scale applications. Medium resolution, freely available data, such as Landsat 8 imagery, provide a more practical option but satisfactory accuracy must be demonstrated for the research problem of terrace identification. Note

that medium-resolution images have been used in many previous large-scale agricultural land cover classification studies (Massey et al., 2018; Phalke and Özdoğan, 2018).

Although the amount of data decreases when conducting classification at medium resolution compared with high resolution images, remote sensing-based land cover mapping at the scale of a large country like China still requires considerable data storage and processing capabilities, which is a major challenge for software packages. The Google Earth Engine (GEE)

platform provides a multi-petabyte analysis-ready data catalog and high-performance, intrinsically parallel computation service (Gorelick et al., 2017). Thus, the handling of large geospatial datasets and access to substantial computational resources can be easily met on GEE, improving the efficiency of geospatial analysis. GEE has been used in the large-scale mapping of land cover and land use. For example, Chen et al. (2017a) mapped the mangrove forests of China, Xiong et al. (2017) mapped cropland of Africa using the GEE, and Liu et al. (2018) produced a time series urban land map at a global

scale based on GEE. These studies have revealed that GEE can efficiently handle various classifications at the national, continental, and global scale.

In the following, we produce a 30 m resolution terrace map for China by a supervised pixel-based classification using the GEE platform, we provide an uncertainty analysis for the mapping results, evaluate the terrace area, and identify the key features for terrace mapping.

## 2 Methods

Our mapping approach included the following steps: (1) data preprocessing, (2) feature calculation, (3) sample collection, (4) classification implementation, (5) postprocessing, (6) accuracy evaluation. The entire mapping procedure was performed on the GEE platform (see Fig. 1). Details for each step of the mapping procedure will be provided in the following sections.

### 2.1 Data and preprocessing

In our study, the following data were mainly used. Their detailed information is summarized in Table 1.

#### 2.1.1 Landsat 8 surface reflectance (SR) imagery

We selected the 30 m Landsat 8 SR product accessible on GEE. This product has been atmospherically corrected and includes a cloud, shadow, water, and snow mask band produced using CFMASK, as well as a per-pixel saturation mask band (USGS, 2018). All scenes covering China acquired in 2018 were selected in our study. After obtaining the 10196 images, we removed the clouds in each image based on the cloud mask band (the pixel QA band) of Landsat 8 SR data. These cloud-free images in 2018 can cover 99.889% of the cropland area as defined by the GlobeLand30 (Chen et al., 2017b). In the provinces with an uncovered cropland area larger than 10 km$^2$, all scenes in 2017 preprocessed as above were supplemented to achieve 99.998% of coverage of cropland areas across China.

#### 2.1.2 Shuttle Radar Topography Mission digital elevation model (SRTM DEM) data

Topographic features being the most distinctive features of terraces, it is crucial to consider them for terrace identification. We selected the 30 m SRTM DEM data for characterizing topographic features. SRTM is an international research effort that obtained digital elevation models on a near-global scale (Farr et al., 2007). The product at 1 arc-second (30 m) resolution is available on GEE, which has undergone a void-filling process using open-source data (ASTER GDEM2, GMTED2010, and NED) (USGS, 2015). Note that the acquisition time of SRTM DEM data is 2000, and we suppose the terrain changes little in decades.

#### 2.1.3 GlobeLand30

To improve terrace mapping efficiency, we limited the classification to cropland area of GlobeLand30, a well-established and widely used source of land cover information, generated by integration of pixel-based and objected-based methods with knowledge (POK) using multi-source data (Chen et al., 2015). Its accuracy for cultivated land type is above 80% (Chen et al., 2017b). The latest version of this land cover dataset is for 2010, which does not correspond to our classification year (2018). However, according to national survey statistics, the cropland area in China changed little in recent years. We thus accepted the use of 2010 data as a mask for our 2018 classification. GlobeLand30 was first downloaded from the website (www.globeland30.org) and then uploaded into GEE.

### 2.1.4 Google Earth images

Google Earth images on GEE were used as auxiliary data for samples collection. This dataset is a composited product combining multiple sets of satellite imagery, which are provided by different commercial image providers or government agencies at different zoom level (Potere, 2008). Its highest resolution can reach less than 1 m. With more detailed information (e.g., texture) provided by the high-resolution Google Earth images, we can visually distinguish the samples more accurately.

## 2.2 Feature calculation

In total 39 features were calculated from the Landsat 8 SR images and the SRTM DEM data, including the 25th, 50th, and 75th percentiles of seven bands and four indexes, as well as six topographic factors (Table 2). These features will be used to distinguish the terraces and non-terraces.

Time series data play an important role in land cover classification, especially for cropland classification, since they can describe phenological traits (Jia et al., 2014; Knight et al., 2006; Matton et al., 2015). However, taking numerous time-series Landsat SR imagery as input for classifiers can lead to performance deterioration due to Hughes phenomenon (i.e. the "curve of dimensionality", accuracy decreases as dimensions increase) (Hughes, 1968). To reduce the dimension of temporal-spectral features, we extracted the percentiles of six optical bands (Blue, Green, Red, NIR, SWIR1, and SWIR2) and one thermal infrared band (TIRS2) from the Landsat 8 time-series images during the whole year. Four important indices in land cover classification were then calculated for each Landsat image, the Normalized Difference Vegetation Index (NDVI), the Normalized Difference Building Index (NDBI), the Bare Soil Index (BSI), and the Modified Normalized Difference Water Index (MNDWI). We then calculated the percentiles of each index, and those percentiles were treated as input features for classification. The above 33 features help to identify the terraces with unique spectral characteristics.

Topographic factors are indispensable in distinguishing terraces from flat crop fields, terraces usually having a steep slope and rough terrain conditions. Therefore, we computed six commonly used topographic factors from SRTM DEM for our classification: elevation, slope, the slope of slope (SOS), roughness (R), slope shape (P), and relief (RF) (Tang et al., 2016).

## 2.3 Training and test samples collection

Due to China's large territory, a large number of training samples are needed to ensure classification accuracy if taking the usual sample collection strategy (i.e. uniform sampling). To improve the sample collection efficiency, we designed a new strategy, which collected national, regional, and local samples in the cropland area from GlobeLand30. Through reusing the national and regional samples, smaller total sample size was required and the workload of sampling was minimized effectively. Since different geomorphologic regions have different typical terrace/non-terrace types, we first collected 801 representative (those are easy to be visually interpreted) samples as national samples in each geomorphologic region (defined in Cheng et al. (2018)) with a large proportion of cropland or typical terrace types (i.e., eastern hilly plains region,

southeastern low-middle mountains region, north China and Inner Mongolia eastern-central mountains and plateaus region, southwestern middle and low mountains, plateaus and basins region). These samples were used for training general classification rules and identifying terraces with typical features. To classify terraces with local or confusing features, a total of 3989 local samples were added to each province to the areas with poor classification visual effect according to the initial classification results. Cropland in some provinces has similar features, such as in Heilongjiang, Inner Mongolia, Liaoning, Jilin, Tibet, and Xinjiang. Since these provinces have a large area, setting regional samples can reduce the workload of sample collection greatly. Therefore, 54 samples in Heilongjiang were taken as regional samples and added to the sample sets of the remaining five provinces respectively. Finally, each province has 407-609 terrace samples and 394-589 non-terrace samples. All 4790 training samples (2151 terrace samples and 2639 non-terrace samples) were collected by visual interpretation of Landsat images, SRTM DEM data, cropland extent data extracted from GlobeLand30, and Google Earth images.

All test samples were randomly generated in a hexagonal grid (Icosahedral Snyder Equal Area Discrete Global Grid (ISEA DGG)) created using the DGGRID software (Sahr, 2019). Based on this strategy, the study area was split into 1460 equal-area regions (Fig. 2). Accurate precision evaluation requires sufficient samples for both terrace and non-terrace; however, the randomly generated terrace samples were insufficient due to their small percentage. To increase the number of terrace samples, we took the following strategy. Four samples were first generated randomly in each hexagon of China. Then, ten random samples were supplemented into each hexagon that contained terrace samples or that surrounded hexagons with terrace samples. Finally, ten samples were randomly added again to the hexagons with terrace samples. The sample interpretation was based on Google Earth images, Landsat images, and SRTM DEM data. We referred to Zhao et al. (2014) to conduct the interpretation and quality control. The samples were double-checked to ensure reliability. A total of 11333 collected samples were acquired within the study area, of which 1092 samples were interpreted as terrace, and 9783 samples as non-terrace. The remaining 458 samples were uncertain or seriously mixed, and were excluded. The terrace test sample is zero in 12 provinces (Beijing, Hainan, Heilongjiang, Hongkong, Jilin, Jiangsu, Macao, Shanghai, Taiwan, Tianjin, Tibet, and Xinjiang), while the terrace/non-terrace test samples are insufficient (N<10 for either terrace or non-terrace) in 14 provinces (Liaoning, Zhejiang, and the above 12 provinces). Thus, terrace area of the 14 provinces was not analyzed in Section 3.3 and accuracy of the 12 provinces was not evaluated in Section 3.4.3.

### 2.4 Terrace/non-terrace classification

We mapped the terraces over entire China using the GEE platform. Random forest was chosen as the classifier for its high precision, efficiency, and stability. It is consisted of multiple decision trees, all of which perform classification separately and vote for the final results. During the training process of decision tree, each tree node is split based on the most contributing feature among the randomly selected input features of the training sample subset. After training, each decision tree judges the pixel class according to the established tree rules (Breiman, 2001; Gislason et al., 2006). The two critical parameters of the random forest classifier, the number of decision trees and the number of variables per split, were set to 200

and the rounded square root of the feature number (39), respectively. The classification process comprised several steps. First, the cropland extent data was used to mask non-cropland area which occupied 80% of the whole study area. As a result, the classification efficiency was improved by at least 80% through masking of non-cropland areas. Second, national, regional, and local training samples were merged to train the classifier of each province and each random forest model was applied to classify the corresponding province. Since terraces are relatively rare and the characteristics of various terraces are quite different, performing the classification separately for each province can obtain better classification results. Finally, we mosaicked all the province maps into a complete terrace map for China.

## 2.5 Postprocessing

### 2.5.1 Mode filtering

In pixel-based classification, different objects with the same spectrum or the same objects with a different spectrum can cause a salt and pepper effect (i.e., impulse noise, presenting as image speckles). Because of the irregular shape and fragmented patches of the terrace, the salt and pepper effect was serious in terrace mapping. Considering that terraces are usually small in size, we used a mode filter with a circle kernel of 1.5 pixels radius to conduct spatial filtering.

### 2.5.2 Sieving

By checking the mapping result, there are still lots of patches with small size that are misclassified. Therefore, sieving was used to remove the noises after the mode filtering. We first sieved the small terrace patches and then the small non-terrace patches. The sieving threshold was set to 10 pixels in this research. Namely, terrace/non-terrace patches with an area of 10 pixels (about 9000 m$^2$) or less were sieved.

## 3 Results and discussions

### 3.1 Mapping result

Figure 3 shows the spatial distribution of China's terraces in 2018. The terraces are mainly concentrated in the plateaus, hills, and basins of China. They are most densely distributed in the Sichuan basin and the Loess Plateau (mainly in Shanxi, Shaanxi, Gansu, Qinghai, and Ningxia provinces), followed by the Yunnan-Guizhou Plateau. Terraces are also widely distributed in the central and southeast hills. The distribution of terraces is closely related to the topography; more terraces are built in regions with uneven/steep terrain to prevent the water runoff and soil erosion. In contrast, there are few terraces on the four great plains of China (i.e., the Northeast Plain, the North China Plain, the Middle-Lower Yangtze Plain, and the Guanzhong Plain). Last, terraces are rare in the Qinghai-Tibet Plateau, the Inner Mongolian Plateau, and Xinjiang, which have small cropland areas.

Visually, our terrace map realistically represents most of the classified areas. Figure 4a–c shows three selected cases of well-known terracing practice in China. Although they were in different provinces and different geographic environments, and their types or appearances varied a lot, they were all correctly classified. The mapping results showed good visual correspondence and coincided well with the Landsat 8 images. Generally, terraces like these with typical features were accurately identified in our map.

## 3.2 Accuracy assessment

Our terrace map was first validated using the 10875 collected test samples mentioned in section 2.3. A confusion matrix was calculated for the classification map to evaluate its accuracy (Table 3). The terrace map had an overall accuracy (OA) of over 94%. The producer's accuracy (PA, also referred to as "1-omission error") of the terrace class was 79.945%, whereas the user's accuracy (UA, also referred to as "1-commission error") of the terrace class was 71.149%, indicating that our results typically overestimated the quantity of terraces. The non-terrace class had both UA and PA over 96%. We also computed the kappa coefficient (Cohen, 1960) to measure the overall classification performance, which was 0.72, indicating that the classification performs well generally.

By visually checking the misclassified samples, we found that most sources of commission errors were sloping cropland, non-crop vegetation (e.g., forests, shrubs, and grasses) in the cropland extent data, and some objects near or on the terrace. The omission errors were mainly caused by the terraces that were not in the cropland extent data, low slope terraces, and small patch terraces (terrace with an area of 10 pixels (about 9000 m$^2$) or less).

We further used another test sample set (N=301) collected by field survey and literature study. Those samples were double checked by visually interpretation of Google Earth images, Landsat images, and SRTM DEM data. The accuracy evaluation result using the 301 test samples of known terraces (Table 4) was numerically similar to the above result using the 10875 random test samples (Table 3). The Chi-square tests (Mantel, 1963) were carried out for the two PAs and UAs of terrace class respectively to further prove the similarity quantitatively. The P-values of both tests were greater than 0.05, indicating there was no statistically significant difference between the terrace accuracies using the two test sample sets. The similarity proved the high accuracy of our terrace map again and confirmed that our random test samples and accuracy evaluation result were reliable. Note that the 301 samples were just used here for accuracy assessment verification, and were not used in the following analysis.

## 3.3 Area estimation

For each province with sufficient test samples (N≥10 for both terrace and non-terrace), we used three methods to estimate the terrace area based on Yu et al. (2017): (1) pixel counting (PC), (2) sample proportion (SP), and (3) error matrix-based model-assisted estimation (EM). The PC method estimates the terrace area by summing the area of pixels classified into terrace. The SP method estimates the terrace area by multiplying the terrace test samples proportion by the total area. The EM method (see Olofsson et al. (2014) for detailed calculation procedures) combines the terrace map and test samples,

which uses the samples proportion to adjust the area estimated by the PC method. Comparisons of the areas estimated by the three methods are shown in Fig. 5.

According to the terrace area at the provincial level, the provinces with the largest terrace extent in China were Sichuan, followed by Yunnan and Gansu, whereas the terrace area was smallest in Guangdong. Guizhou, Chongqing, Yunnan, and Sichuan had the highest proportion of terrace, with more than 70% of the cropland having been terraced. The provinces with large terrace percentages were mainly located in southwest China or the Loess Plateau. In the evaluated 20 provinces, except for Inner Mongolia, terrace proportion was also relatively low in a few eastern and southern provinces.

For most provinces, terrace area estimates from the three methods were similar. This consistency can also be regarded as a robust test for the mapping accuracy. However, the area calculated by the SP method in Sichuan, Gansu, and Qinghai was abnormally high. The outliers are mainly caused by the test sample generation strategy, which leads to a large proportion of terrace test samples in these provinces. Compared with the PC and SP method, the EM method provides the confidence intervals for area estimates. In addition to the several provinces with a large proportion of terraces, the confidence intervals

were large in Inner Mongolia because of the low accuracy.

Shanghai and Macau are the only two provinces in China have zero terrace area by PC method in our map. Terrace area for other 14 provinces (Beijing, Hainan, Heilongjiang, Hongkong, Jilin, Jiangsu, Liaoning, Macao, Shanghai, Taiwan, Tianjin, Tibet, Xinjiang, and Zhejiang) has not been analyzed due to insufficient test samples to estimate the uncertainty.

As for the total area of terraces in China for 2018, it was estimated to be 53.55 Mha by the PC method, accounting for 26.43%

of China's cropland area and 5.58% of China's land area (about 960 Mha). And the EM method showed that the total terrace area was 58.46±2.99 Mha, i.e., 28.85%±1.48% of China's cropland area and 6.09%±0.31% of China's land area.

## 3.4 Uncertainty analysis

In addition to the terrace map itself, the associated map quality analysis is also important, which can provide valuable information for users to select data and indicate the limitations and future improvements. In the following subsections, the

270 mapping uncertainty was assessed using the traditional accuracy evaluation method at the sampling unit level, terrain level, and province level, i.e., calculating the accuracy of different sampling units, terrain, and provinces. Then, the probability of terrace versus non-terrace was calculated to evaluate the uncertainty at the pixel level.

### 3.4.1 Sampling unit-level uncertainty

Sampling unit-level accuracy provides information about the spatial variation of uncertainty for the map (Fig. 6). The

275 accuracy within each hexagon was evaluated based on the test samples that fall inside it. The OA was above 75% in most hexagons (Fig. 6a). Unlike the OA, which only considers the samples correctly classified, kappa also takes the commission and omission errors into account. Figure 6b indicates that the poorly classified hexagons were mostly concentrated in the northeastern, southern, and central-eastern regions. According to Fig. 6c, the hexagons with the highest PA for terrace class were mainly located in southwest China and the Loess Plateau. Some regions in the middle and south of China had low PA,

where some low slope terraces were difficult to identify because they have similar features with flat cropland (non-terrace cropland with a slope of <5°). Moreover, misclassification between terraces and strip planting led to larger omission errors in the northeastern region. The distribution pattern of UA for terrace class (Fig. 6d) demonstrates that most regions had apparent overestimation except the northwestern region. In southwestern China and the east of the Loess Plateau, sloping cropland, and non-cropland vegetation types (e.g., forests, shrubs, and grasses) were the main commission errors. In northeastern China, confusion between terraces and strip planting also resulted in the poor UA. In central China, flat cropland led to the major overestimation. In South China, the low UA was mainly caused by both the sloping cropland and the flat cropland. However, it should be noted that the western hexagons generally contained more terrace samples, whereas some eastern hexagons or hexagons on edge had very few terrace samples (N<2), which also could have influenced the result.

### 3.4.2 Terrain-level uncertainty

As crucial factors in terrace/non-terrace classification, terrain has a major impact on classification accuracy. Terraces with obvious terrain features, such as specific slope and elevation, are easier to identify. The terrain-related uncertainty of our data is shown in Fig. 7. The OA showed little difference between various terrain types and all exceeded 90%. In contrast, kappa was more suited to judging the poorly classified terrain zones. Kappa was the lowest for regions with an elevation less than 200 m. Accordingly, these regions also had poor PA and UA (Fig. 7a). Terraces in these regions were difficult to identify because of their similar topographic features with flat cropland, and thus were easily confused with them. As for the slope (Fig. 7b), kappa increased first and then decreased. The confusion between low slope terraces and flat cropland caused the most omission errors and commission errors in the low slope regions. On the other hand, the relatively low PA of the high slope regions was mainly caused by terraces not in the cropland extent data, whereas the poor UA was mainly caused by sloping cropland and non-crop vegetation types.

### 3.4.3 Province-level uncertainty

Since the terrace map of different provinces was classified with different training samples, we evaluated the accuracy of 22 provinces out of 34 in China, excluding 12 provinces (Beijing, Hainan, Heilongjiang, Hongkong, Jilin, Jiangsu, Macao, Shanghai, Taiwan, Tianjin, Tibet, and Xinjiang) without terrace test samples (Fig. 8). The accuracies for Liaoning and Zhejiang may be biased because of the insufficient number of test samples (N<10 for either terrace or non-terrace), which are indicated by "*" in Fig. 8. To avoid biases, these provinces were excluded from further analyses in this study.

Fifteen out of twenty-two assessed provinces had kappa greater than 0.6. Three northwestern provinces (Qinghai, Gansu, and Ningxia) achieved the highest kappa values, the wide terrace there were easier to identify. Shandong and Fujian also had relatively high kappa values, which are 0.78 and 0.75, respectively. Kappa values in the provinces of southwestern China and the Loess Plateau reached 0.7 except Shaanxi, Shanxi, and Yunnan, where the relatively lower accuracies were caused by the confusion with sloping cropland and non-crop vegetation types. Two of the lowest kappa values were for Liaoning and Zhejiang, and were thus likely also to be influenced by the insufficient test samples. Besides, kappa values of

Guangdong, Inner Mongolia, Guangxi, and Hebei were also low. In these poorly classified provinces, commission errors predominated in Guangxi and Hebei, and both omission errors and commission errors were numerous in Zhejiang, Liaoning, Guangdong, and Inner Mongolia. Moreover, PA was higher than UA in most provinces, indicating overestimation was more serious than underestimation in general. Anhui was an exception, where omission errors were much higher because of the large proportion of low slope terraces.

### 3.4.4 Pixel-level uncertainty

The random forest classifier can provide information on the pixel-level uncertainty based on the classification probability of the pixel being terraced or not. The classification uncertainty of a pixel was calculated by $1-P_{max}$, where $P_{max}$ was the maximum probability of being classified into terrace class and non-terrace class (Loosvelt et al., 2012). As shown in Fig. 9, the regions with the lowest uncertainty were the plains of China. Besides, the Loess Plateau and southwestern China with large area of terraces also had relatively low uncertainty. The terraces and non-terraces there had different features in relation to each other. Thus, the classification model performed well. In contrast, the pixel-level uncertainty was higher in the southern, central-eastern and part of the northeastern region. Here, the terraces had heterogeneous features and some features are similar with some non-terraces, leading to the classification uncertainty. These are important sources of uncertainty that need to be addressed in future research. Although the pixel-level uncertainty results generally corresponded well with the above general accuracy evaluation results, there were some differences. In some regions with high accuracy, the pixel-level uncertainty can also be relatively high (such as northeastern Inner Mogolia, Jianghan Plain), and some low accuracy regions can also have low pixel-level uncertainty (such as some misclassified area in southeastern China and the Loess Plateau), which may be influenced by the small number or confused features of training samples in these regions.

### 3.5 Feature importance evaluation

Identification of the most important features for terrace mapping is significant for further studies. The feature importance can be evaluated by the decrease of some impurity measures, such as Gini index (Louppe et al., 2013). It is related to the training samples, in which different training sample sets used to train the random forest model can obtain different importance results. To achieve a stable and reliable feature importance ranking, the importance values based on the various training sample sets for different provinces were first acquired, and then their mean and standard deviation were calculated (Fig. 10).

According to Fig. 10a, most provinces show a similar feature importance pattern under our sample collection strategy. Elevation and slope had the greatest contribution to the final decision in nearly all the provinces, confirming that topographic characteristics are crucial factors for identifying terraces. In general, there were distinguishable differences between the elevation range or slope range of the terraces and those of the non-terraces. The importance of the two features was relatively low in the provinces with lots of sloping cropland or low slope terraces. In addition to elevation and slope information, multi-temporal spectral features are necessary. For most provinces, NDVI_25th, TIRS2_25th, and BSI_75th were essential features in terrace mapping. However, there were several optical band percentiles (e.g., SWIR1_50th and SWIR2_50th)

showing low importance almost in all provinces. These variables contribute little to terrace/non-terrace classification and can

be removed in a future study to reduce feature dimensions and improve classification efficiency. Moreover, some features such as RF and Blue_25th displayed different importance patterns in various provinces. Therefore, the decision on whether to include these features depends on the study area.

Figure 10b shows a summary of the feature importance of terrace/non-terrace classification. Elevation ranked first, followed by slope, and their importance values were more than 20% of the third-place feature (NDVI_25th). The top ten features

included three topographic factors (elevation, slope, and SOS), three percentiles of indexes (NDVI_25th, BSI_75th, and NDBI_25th) and four percentiles of bands (TIRS2_25th, SWIR2_25th, TIRS2_50th, and TIRS2_75th), indicating that multi-source and multi-temporal features are indispensable; the integration can increase the separability and guarantee classification accuracy across different terrace landscapes. In contrast, the categories of the least important features were more consistent. Eight of the last ten features were all percentiles of bands. For the standard deviation, in addition to

elevation and slope, RF and Blue_25th also had high values, implying the importance values of these features varied greatly among different provinces.

### 3.6 Limitations and directions for future research

Our terrace map achieved satisfactory accuracy and good visual effect as a whole over all China and different provinces, indicating that the terraces and non-terraces can be distinguished using 30 m resolution satellite imagery and supervised

pixel-based classification. Figure 11a, b present two accurately identified cases from the terrace map. They were located in Gansu and Sichuan, with lots of terraces. It was easy to recognize them based on the Landsat images. In some other provinces, it was also easy to recognize differences between terraced and non-terraced cropland generally (Fig. 12). In our map, almost all terraces with distinctive features and large patch areas were identified well, suggesting our classification method can deal with them effectively. Here, we also quantitatively testified the rationality and validity of feature selection

and sampling strategy in the study. To further illustrate the usefulness of all the 39 features selected in the study, we took a terraced cropland-dominated province (Guizhou) and a non-terraced cropland-dominated province (Hubei) as examples to train the classification model based on different feature numbers and evaluate the accuracy. According to Fig. 13, OA generally showed an upward and gradually stable trend as the feature number increased in both provinces, the maximum values were reached when using 35 features in Guizhou and 39 features in Hubei, indicating features were not redundant.

Therefore, we applied all features in this study. As for the sampling strategy, to further clarify the effectiveness of the national and regional training samples in the study, we compared the accuracies of classification through only using local training samples and through using national, regional and local training samples in the provinces where both local terrace and local non-terrace training sample size were more than 10 (Anhui, Beijing, Chongqing, Fujian, Guangdong, Guangxi, Hainan, Hebei, Henan, Hubei, Hunan, Inner Mongolia, Jiangsu, Jiangxi, Liaoning, Qinghai, Shaanxi, Shanxi, Sichuan,

Taiwan, Tianjin, Zhejiang). On the whole, adding the national and regional samples increased OA by 6.90% in these provinces, proving our sampling strategy in the study is reliable and can be applied to other large-scale researches. However,

there are still a number of limitations and difficulties in the terrace/non-terrace classification field, which are expected to be explored by future research.

First, difficulties in terrace feature characterization hinder the identification. Terrace types varied across our large-scale study area, causing the spectral and topographic information from different terrace classes to vary considerably. Moreover, some terraces shared similar features with non-terraced cropland. Feature heterogeneity of different terraces and feature similarities of terraces and other land covers increased the difficulties of characterization, thus causing some typical confusion between classes, for instance, terraces and sloping cropland (Fig. 11c), terraces and strip planting (Fig. 11d), low slope terraces and flat cropland (Fig. 11e). In some regions, there are no apparent differences between the above terraces and non-terraces in the extracted spectral features and topographic features. The only difference between them is whether or not there are steps; however, this cannot easily be depicted by the pixel-based classification at 30 m resolution. Steps divide the continuous terrace landscape into an array of regularly sized pixels with similar features, exhibiting strong spatial auto correlation in high-resolution remote sensing images, which can be incorporated into the classification through the computation of texture measures. Several studies on land cover mapping have shown that adding textural variables can increase the mapping accuracy (Johansen et al., 2007; Masjedi et al., 2016; Rodriguez-Galiano et al., 2012). Moreover, some small-scale research has shown the effectiveness of textures in terrace/non-terrace classification (Li et al., 2013; Luo et al., 2020; Zhang et al., 2017). The ability of textures to discriminate different land cover types relates to the image spatial resolution (Chen et al., 2004). High-resolution images are required to extract texture information for terraces. Compared with the pixel-based method, which only uses the direct pixel feature information alone, the object-based method aggregates a group of pixels together as an object and integrates its spatial, textural and contextual information (Liu and Xia, 2010). For some land cover types, object-based classification has proven to be a better approach (Myint et al., 2011; Pande-Chhetri et al., 2017). Terraces are also suitable to be treated as patches. Single pixels of some terraces show no distinct features; however, the large set of characteristics for the entire terrace patch can be identified easily. The object-based classification method has also been testified in some small study areas (Diaz-Varela et al., 2014; Zhao et al., 2017). However, the parameter setting of the method is difficult for large-scale classification because different study areas usually require different parameter values to gain optimal results. So, it can be used as an optimization method in low precision regions. Although automatic classification is prevalent and shows good performance in the classification field, it is still unsuitable for regions with confusing landscape. In such cases, visual interpretation can be used as an assistant method. The combination of digital classification and visual interpretation has been found to increase the accuracy significantly (Raši et al., 2011; Shalaby and Tateishi, 2007). Thus, for regions with features that confuse the automatic classification system in terrace mapping, visual interpretation can be applied to provide more reliable results. In summary, combining texture features from high-resolution images, conducting object-based classification and visual interpretation in poorly-classified regions are worth attempting in future research.

Second, the irregular shape and small patch size of terraces make the classification more difficult. Mixed pixels are common in medium-resolution images because of the spatial resolution limitations and spectral heterogeneity (Fisher, 1997). This

phenomenon is especially severe in terrace/non-terrace classification. The use of 30 m Landsat data in this research ensures that terraces with large areas can be detected. However, with data of this resolution it is not possible to correctly classify fragmented terrace patches (Fig. 11f). Moreover, the proportion of terraces within a pixel can cause overestimation or underestimation. Higher resolution images can generate a more accurate classification. To better capture these small and

irregular terraces, satellite images with a finer resolution are necessary. However, the hard classification (i.e. classification which gives the discrete class of land cover) will still have this problem even if the resolution is higher. Discrete classes cannot represent spatially complex areas well. In recent years, some classification studies have produced continuous fractions of land cover at the pixel scale through unmixing to address subpixel heterogeneity (Atzberger and Rembold, 2013; Deng and Wu, 2013; Xie et al., 2016). Therefore, to tackle the mixed pixel issue, future work should focus on improving the

image resolution or the estimation of subpixel terrace distribution.

In addition to the above difficulties in terrace identification, both the terrain data (SRTM data) and the cropland extent data extracted from GlobeLand30 caused some limitations in the mapping results. On the one hand, the inconsistent year of terrain data and classification led to the uncertainties. Terrain may change due to the land use/cover transformations during 2010-2018. However, in relative to the vertical accuracy of terrain data, most transformations had little impact on terrain.

Even if terrain changed significantly in somewhere during the eight years, the spectral features in 2018 can help with classification. And the satisfactory accuracy of our terrace map also indicated the assumption of little terrain change was acceptable. But there is no doubt that better results can be achieved if high resolution and precision terrain data in 2018 is available. On the other hand, the errors in the cropland extent data propagated into our terrace map. The omitted cropland led to some omission errors in terrace mapping. Besides, the commission errors in the cropland extent data led to some

uncertainties. Some abandoned terraces, terraces used for reforestation or afforestation in the cropland extent data were classified into terraces. And even some non-cropland vegetation not on the terraces and a few other objects near or on the terraces caused part commission errors. Furthermore, the non-correspondence of cropland extent data year and terrace/non-terrace classification year also had an impact on the results. Compared to using a cropland map in 2018, the major limitation of using cropland map in 2010 is that some terraces located in the newly increased cropland area will be omitted. We

quantified the omission caused by the cropland mask using the test samples. Only 119 of the total 1092 terrace test samples located outside the cropland extent of GlobeLand30 2010, indicating the maximum possible omission errors caused by the non-corresponding year was 10.90%. These uncertainties can be avoided by using accurate cropland extent data. However, we had tried several other best cropland extent data at present to limit the classification area, such as cropland of National Land Cover Database for China (NLCD-C) (Zhang et al., 2014) and Global Food Security-support Analysis Data at 30 m

(GFSAD30) (Teluguntla et al., 2017), the results were worse compared to using the GlobeLand30. We had also tried to use the high-accuracy China forest map (Li et al., 2014) to mask the reforestation or afforestation terraces and some misclassified forests, but the accuracy of terrace identification decreased due to the errors in the forest map. Thus, more accurate cropland extent data and forest extent data are urgently needed to deal with this problem.

**4 Data availablity**

The China terrace proportion map at 1 km resolution (calculated from 30 m resolution map) is available to the public at https://doi.org/10.5281/zenodo.3895585 (Cao et al., 2020). The map values indicate the proportion of terrace within 1×1 km grid cell. The China terrace map at 30 m resolution will be available after publication.

The Landsat 8 SR imagery and SRTM DEM data used in this study were available at the GEE platform. The GlobeLand30 data can be downloaded from www.globeland30.org.

**5 Conclusions**

In this study, the first 30 m resolution terrace map of China was developed through supervised pixel-based classification using multi-source, multi-temporal data based on the Google Earth Engine (GEE) platform. The overall accuracy (OA) was 94.73% and kappa was 0.7235, indicating that the supervised pixel-based classification at 30 m resolution was generally feasible for large-scale terrace mapping. The classification achieved the highest accuracy in southwestern China and the

Loess Plateau. The sloping cropland (non-terrace cropland with a slope of ≥5°), low slope terraces, and non-crop vegetation caused the most errors in the mapping results. Zhejiang, Liaoning, Guangdong, Inner Mongolia, and Guangxi were the five provinces with the lowest accuracy. According to our map, terraces are widely distributed in China and most are built in southwestern China and the Loess Plateau. The total terrace area was estimated to be 58.46±2.99 Mha by error matrix-based model-assisted estimation (EM) method, accounting for 28.85%±1.48% of China's cropland area. Feature importance

analysis indicated that elevation and slope were the most important features for terrace identification, but time series spectral features were also necessary to achieve satisfactory results. The intra-class feature heterogeneity, inter-class feature similarity and fragmented patches make terraces challenging to identify. Thus, future research should focus on better characterizing terrace features and recognizing small patch terraces. This novel terracing map can be used for large-scale soil erosion and carbon cycle studies, as well as for assessments of multiple ecosystem services of terracing. The terrace

identification tool can be extended to other regions globally, but will need to be validated to those regions as terrace types can differ significantly.

**Author contribution**

BC and LY conceived the study. All authors (BC, LY, VN, PC, WL, YZ, WW, DC, ZL, PG) contributed to the discussions, writing, and analysis of the manuscript.

**Competing interests**

The authors declare that they have no conflict of interest.

## Acknowledgements

This research has been supported by the National Key R&D Program of China (grant nos. 2019YFC1510003, 2017YFA0604401 and 2019YFA0606601).

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

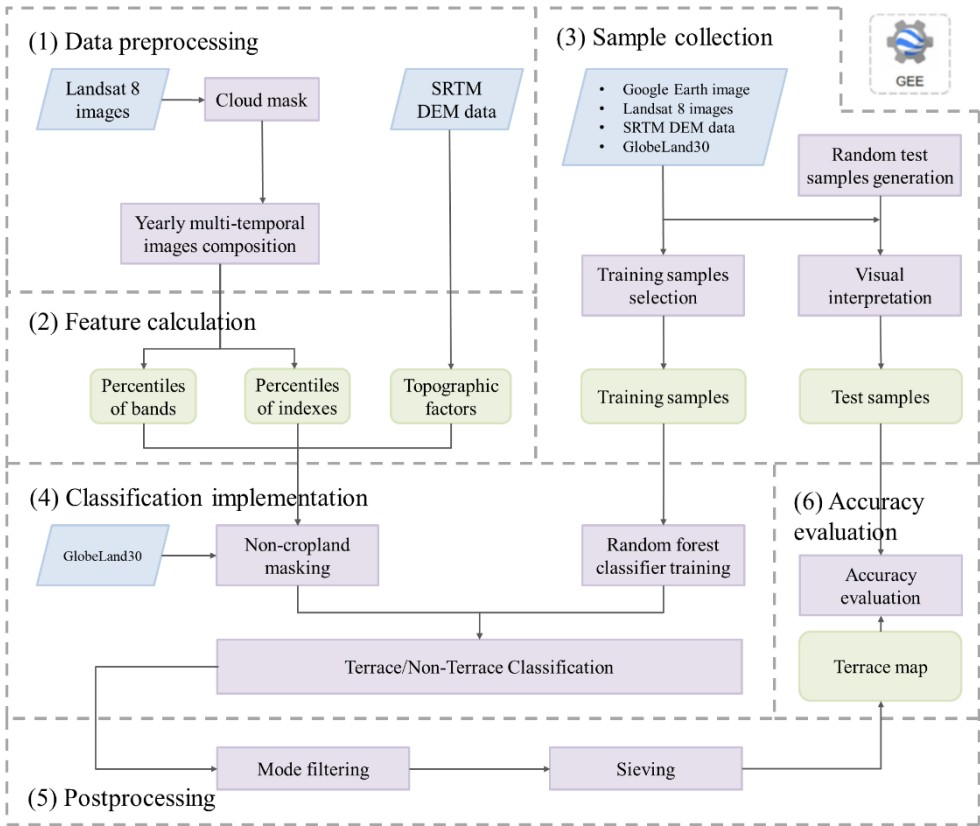

Figure 1: The framework of terrace/non-terrace classification applied in the current study. The whole project was accomplished based on the GEE platform.

**Table 1: Detailed information of data used in this study.**

|  | Landsat 8 surface reflectance (SR) imagery | Shuttle Radar Topography Mission digital elevation model (SRTM DEM) data | GlobeLand30 | Google Earth images |
|---|---|---|---|---|
| Title | USGS Landsat 8 Surface Reflectance Tier 1 | NASA SRTM Digital Elevation 30m | GlobeLand30 | / |
| Version | / | Version 3 | V2010 | / |
| Acquisition time | 2018 | 2000 | 2010 | Most images were accessed in 2018-2019 |
| URL | https://developers.google.com/earth-engine/datasets/catalog/LANDSAT_LC08_C01_T1_SR | https://developers.google.com/earth-engine/datasets/catalog/USGS_SRTMGL1_003 | http://www.globallandcover.com | / |
| Description | This dataset is the surface reflectance from the Landsat 8 OLI/TIRS sensors. It has been atmospherically corrected and includes a cloud, shadow, water, and snow mask band, as well as a per-pixel saturation mask band. | This dataset is an international research effort that obtained DEM on a near-global scale. It has undergone a void-filling process using open-source data (ASTER GDEM2, GMTED2010, and NED). | This dataset is a 30-meter resolution global land cover data product. It is generated by integration of pixel-based and objected-based methods with knowledge (POK) using multi-source data. | This dataset is a composited product combining multiple sets of satellite imagery. The satellite images are provided by different commercial image providers or government agencies at different zoom level. |
| Image sources | / | / | / | CNES / Airbus, Maxar Technologies, Landsat / Copernicus |


**Table 2: Features for terrace/non-terrace classification.**

| Feature | Data source/Algorithm |
|---|---|
| *Percentiles of spectral bands/indices* | *Landsat 8 surface reflectance (SR) imagery* |
| Bands | Landsat 8 SR Band 2—7 (Blue, Green, Red, NIR, SWIR1, SWIR2), Band 11 (TIRS2) |
| Normalized Difference Vegetation Index (NDVI) | $\dfrac{(\text{NIR} - \text{Red})}{(\text{NIR} + \text{Red})}$ |
| Modified normalized Difference Water Index (MNDWI) | $\dfrac{(\text{Green} - \text{SWIR1})}{(\text{Green} + \text{SWIR1})}$ |
| Normalized Difference Building Index (NDBI) | $\dfrac{(\text{SWIR1} - \text{NIR})}{(\text{SWIR1} + \text{NIR})}$ |
| Bold Soil Index (BSI) | $\dfrac{((\text{SWIR1} + \text{Red}) - (\text{Blue} + \text{NIR}))}{((\text{SWIR1} + \text{Red}) + (\text{Blue} + \text{NIR}))}$ |
| *Topographic factors* | *Shuttle Radar Topography Mission digital elevation model (SRTM DEM) data* |
| Elevation | SRTM DEM data |
| Slope | $\dfrac{\text{Elevation change}}{\text{Horizontal distance change}}$ |
| Slope of Slope (SOS) | $\dfrac{\text{Slope change}}{\text{Horizontal distance change}}$ |
| Roughness (R) | $\dfrac{S_{\text{curved surface}}}{S_{\text{plane surface}}}$ |
| Slope shape (P) | $H_{i,j} - \dfrac{\sum_{i=1}^{n} H_i}{n}$ |
| Relief (RF) | $H_{\max} - H_{\min}$ |

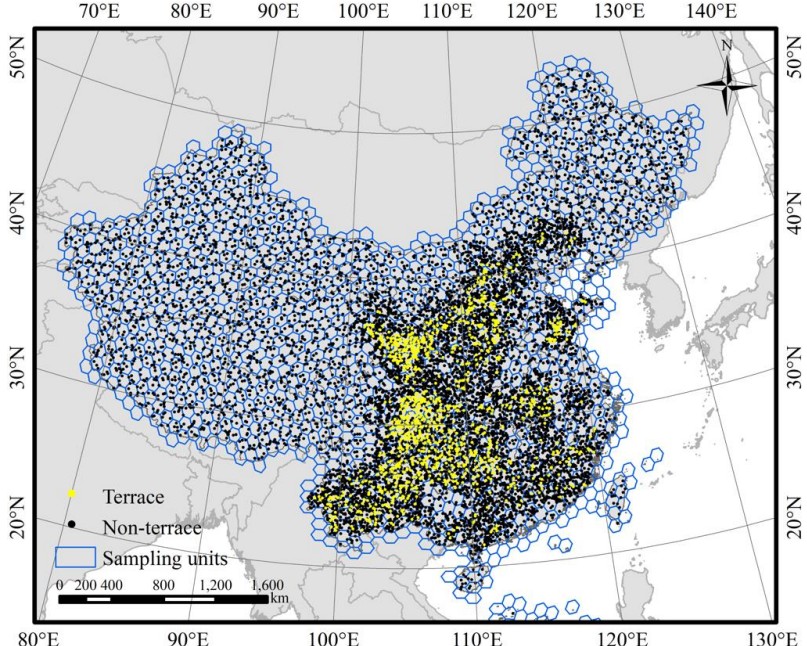


**Figure 2: The spatial distribution of the test samples.**

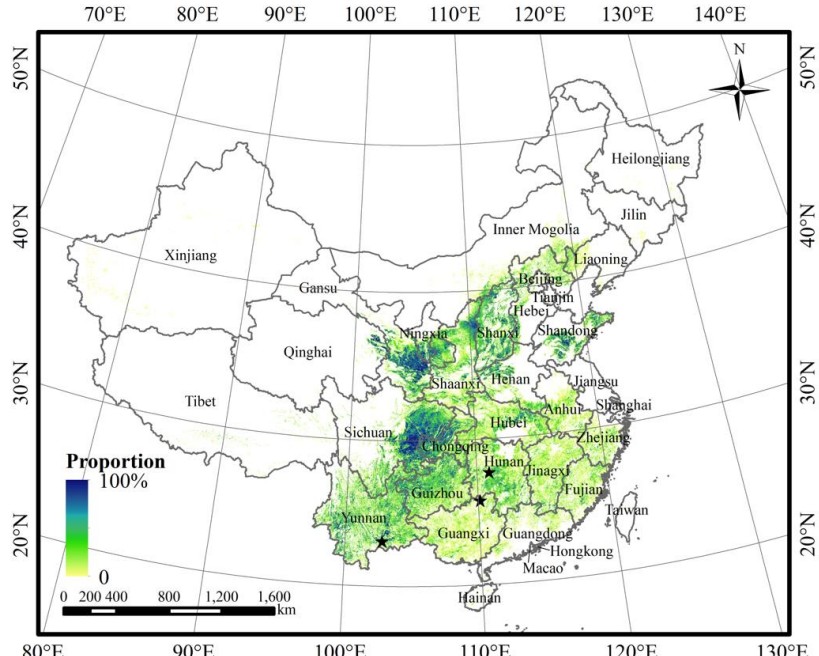

**Figure 3: Terrace distribution across China in 2018. The map values indicate the proportion of terrace within a 1km×1 km grid cell except for Heilongjiang, Liaoning and Xinjiang, where the mapping results are displayed at 5km×5 km resolution for clearer visual effect. Shanghai and Macao are the only two provinces have no terrace in this map. The locations of three well-known terraces shown in Fig. 4 are marked as stars in the terrace map.**


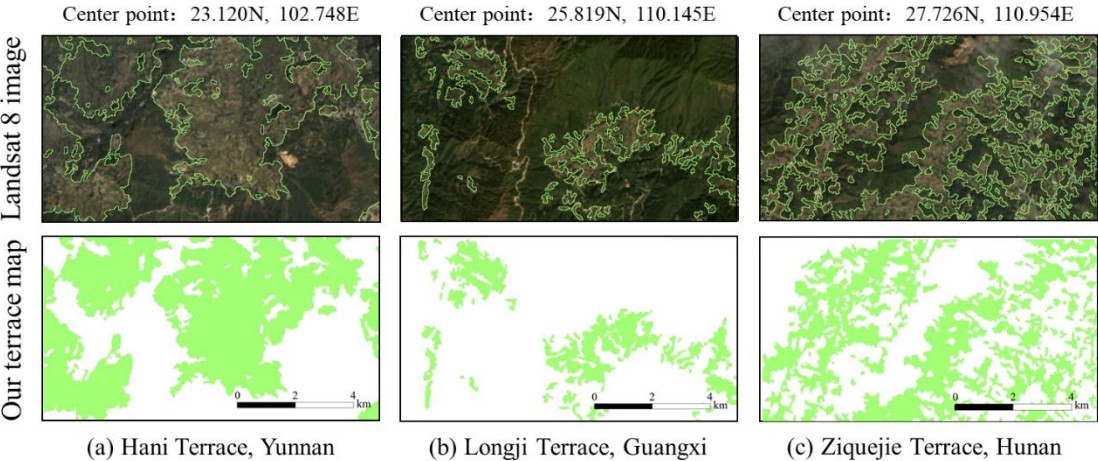

Figure 4: Visual comparison between Landsat 8 images and our terrace mapping results (green represents terrace and white represents non-terrace) for 3 areas with well-known terracing practice: (a)-(c). The locations of these terraces are marked as stars in Fig. 3. The green lines in Landsat 8 images are the corresponding terrace boundaries in our terrace map.

**Table 3: Accuracy assessment of the 30 m terrace map by the 10875 random test samples.**

| Class | Non-terrace | Terrace | PA (%) | UA (%) |
|---|---|---|---|---|
| Non-terrace | 9429 | 354 | 96.381 | 97.730 |
| Terrace | 219 | 873 | 79.945 | 71.149 |
| OA = 94.731% | | Kappa = 0.72353 | | |


**Table 4: Accuracy assessment of the 30 m terrace map by the 301 test samples of known terraces.**

| Class | Non-terrace | Terrace | PA (%) | UA (%) |
|---|---|---|---|---|
| Non-terrace | 201 | 18 | 91.781 | 92.202 |
| Terrace | 17 | 65 | 79.268 | 78.313 |
| OA = 88.372% | | Kappa = 0.70779 | | |

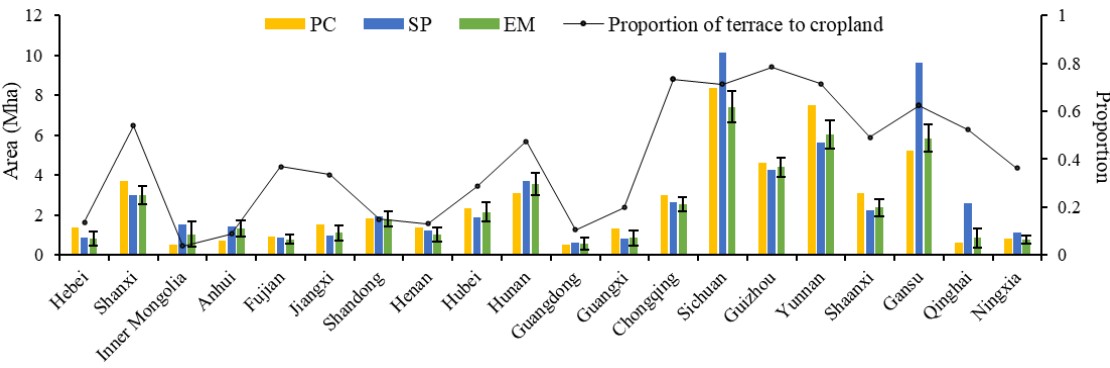

**Figure 5: The terrace area and proportion in different provinces of China. The error bars indicate a 95% confidence interval. The proportion of terrace to cropland (cropland in GlobeLand30) was calculated by the pixel counting method.**


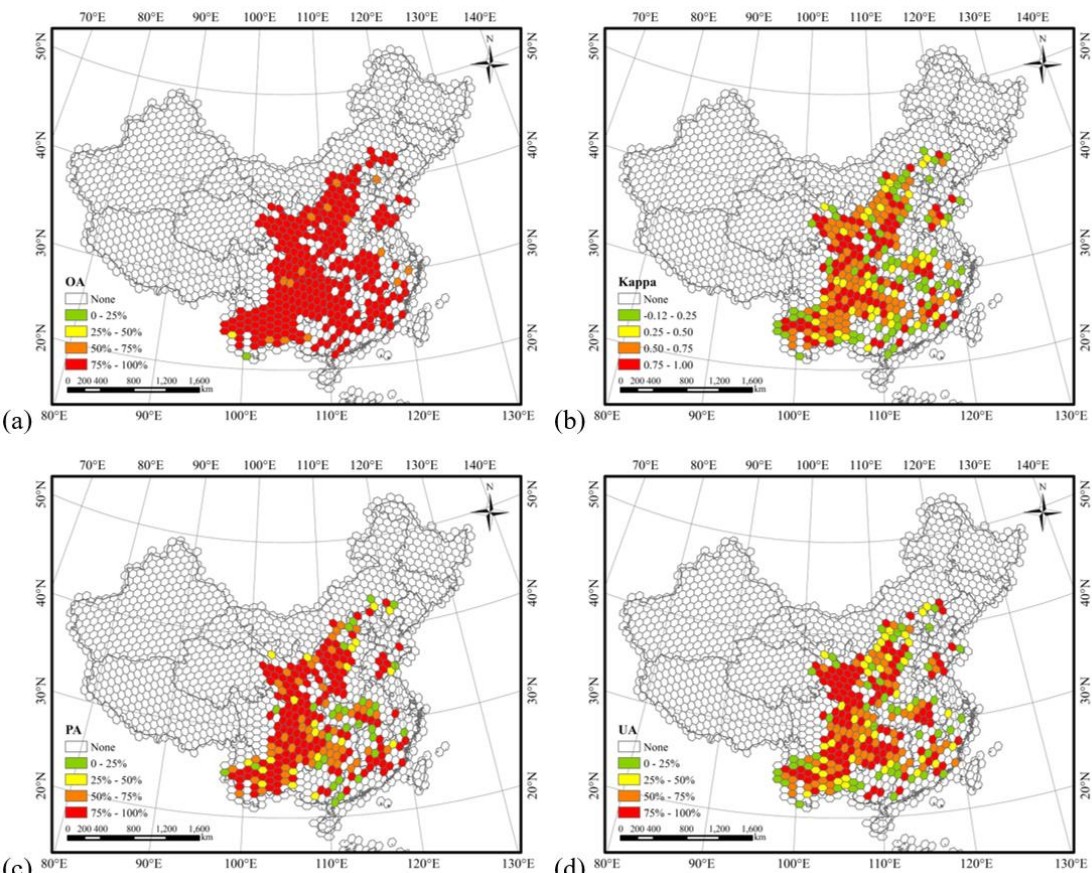

**Figure 6: Spatial variation of (a) OA (only calculated for hexagons with terrace test samples or with non-terrace test samples classified into terrace class), (b) kappa, (c) PA of terrace class, and (d) UA of terrace class.**

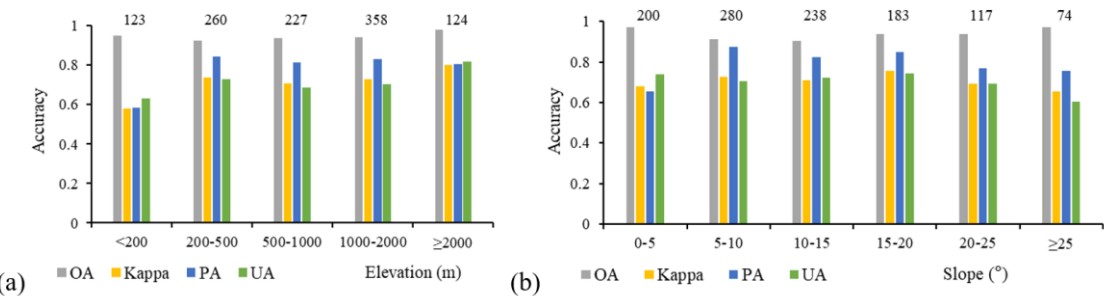

**Figure 7: OA, kappa, PA of terrace class and UA of terrace class for (a) different elevation, and (b) different slope. The numbers marked in the figure represent the terrace sample size within the corresponding elevation or slope range. Two test samples not covered by SRTM DEM data were not used in the terrain-related accuracy evaluation.**


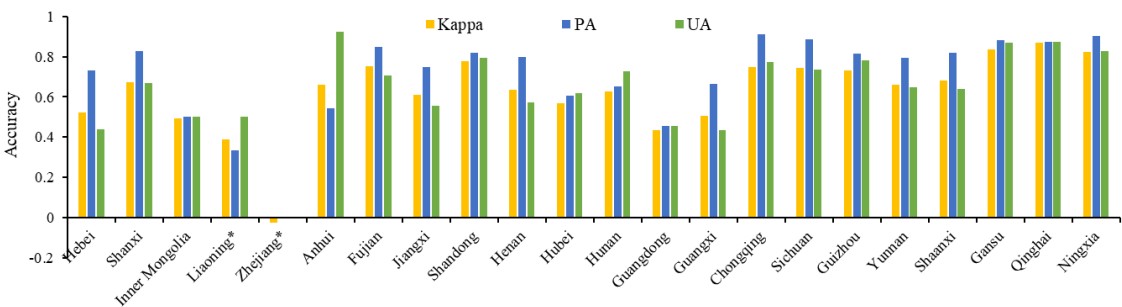

**Figure 8: Kappa, PA of terrace class and UA of terrace class for different provinces.**


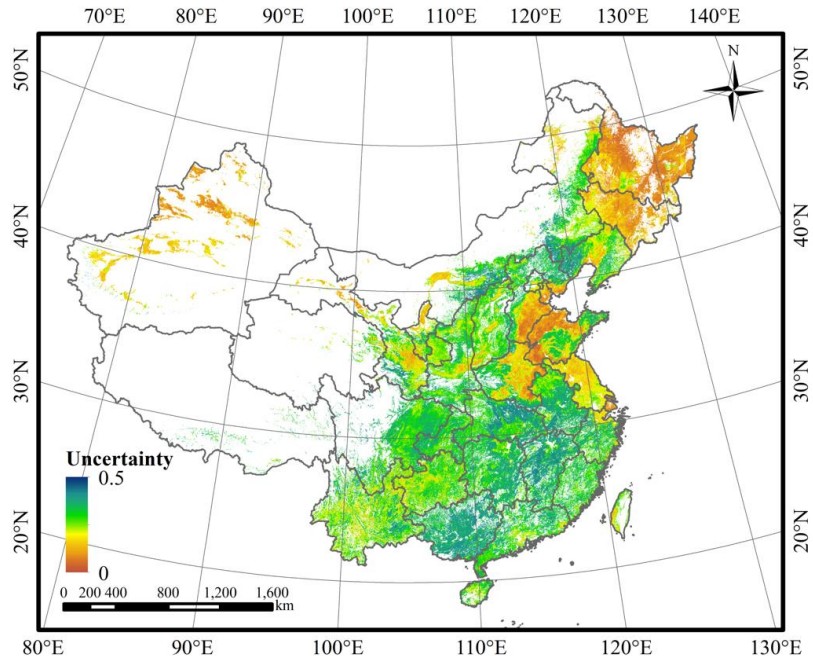

**Figure 9: Pixel-level uncertainty of terrace mapping result. The uncertainty map was resampled to 1km × 1 km spatial resolution.**

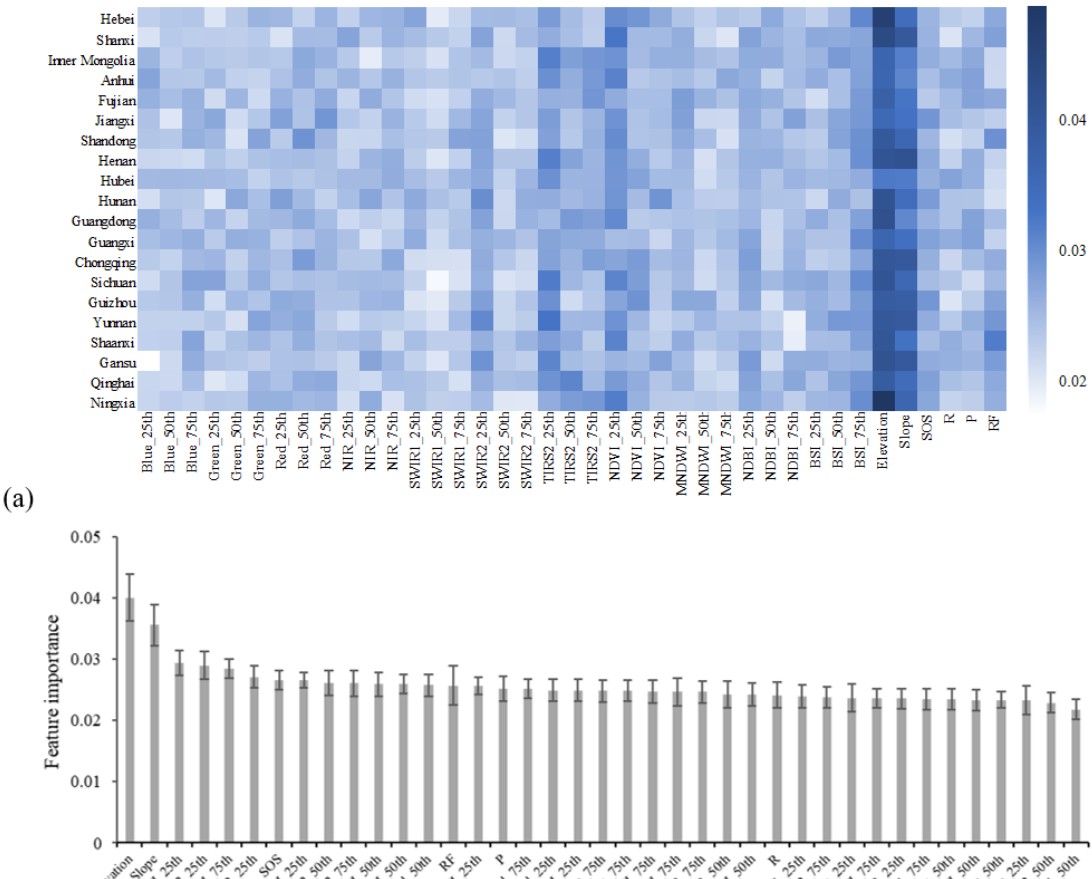

(a)

(b)

**Figure 10: (a) Feature importance value for different provinces, and (b) average feature importance of provinces. The error bars indicate the standard deviation. The variable names were explained in Table 5.**

**Table 5: Explanation of variables in Fig. 10 of Section 3.5.**

| Variable | Explanation |
| --- | --- |
| Blue_25th | The 25th percentile of surface reflectance of Blue band. |
| Blue_50th | The 50th percentile of surface reflectance of Blue band. |
| Blue_75th | The 75th percentile of surface reflectance of Blue band. |
| Green_25th | The 25th percentile of surface reflectance of Green band. |
| Green_50th | The 50th percentile of surface reflectance of Green band. |
| Green_75th | The 75th percentile of surface reflectance of Green band. |
| Red_25th | The 25th percentile of surface reflectance of Red band. |
| Red_50th | The 50th percentile of surface reflectance of Red band. |
| Red_75th | The 75th percentile of surface reflectance of Red band. |
| NIR_25th | The 25th percentile of surface reflectance of NIR band. |
| NIR_50th | The 50th percentile of surface reflectance of NIR band. |
| NIR_75th | The 75th percentile of surface reflectance of NIR band. |
| SWIR1_25th | The 25th percentile of surface reflectance of SWIR1 band. |
| SWIR1_50th | The 50th percentile of surface reflectance of SWIR1 band. |
| SWIR1_75th | The 75th percentile of surface reflectance of SWIR1 band. |
| SWIR2_25th | The 25th percentile of surface reflectance of SWIR2 band. |
| SWIR2_50th | The 50th percentile of surface reflectance of SWIR2 band. |
| SWIR2_75th | The 75th percentile of surface reflectance of SWIR2 band. |
| TIRS2_25th | The 25th percentile of surface reflectance of TIRS2 band. |
| TIRS2_50th | The 50th percentile of surface reflectance of TIRS2 band. |
| TIRS2_75th | The 75th percentile of surface reflectance of TIRS2 band. |
| NDVI_p25 | The 25th percentile of Normalized Difference Vegetation Index. |
| NDVI_p50 | The 50th percentile of Normalized Difference Vegetation Index. |
| NDVI_p75 | The 75th percentile of Normalized Difference Vegetation Index. |
| MNDWI_p25 | The 25th percentile of Modified Normalized Difference Water Index. |
| MNDWI_p50 | The 50th percentile of Modified Normalized Difference Water Index. |
| MNDWI_p75 | The 75th percentile of Modified Normalized Difference Water Index. |
| NDBI_p25 | The 25th percentile of Normalized Difference Building Index. |
| NDBI_p50 | The 50th percentile of Normalized Difference Building Index. |
| NDBI_p75 | The 75th percentile of Normalized Difference Building Index. |
| BSI_p25 | The 25th percentile of Bold Soil Index. |
| BSI_p50 | The 50th percentile of Bold Soil Index. |
| BSI_p75 | The 75th percentile of Bold Soil Index. |
| Elevation | Elevation. |
| Slope | Slope. |
| SOS | Slope of slope. |
| R | Roughness. |
| P | Slope shape. |
| RF | Relief. |

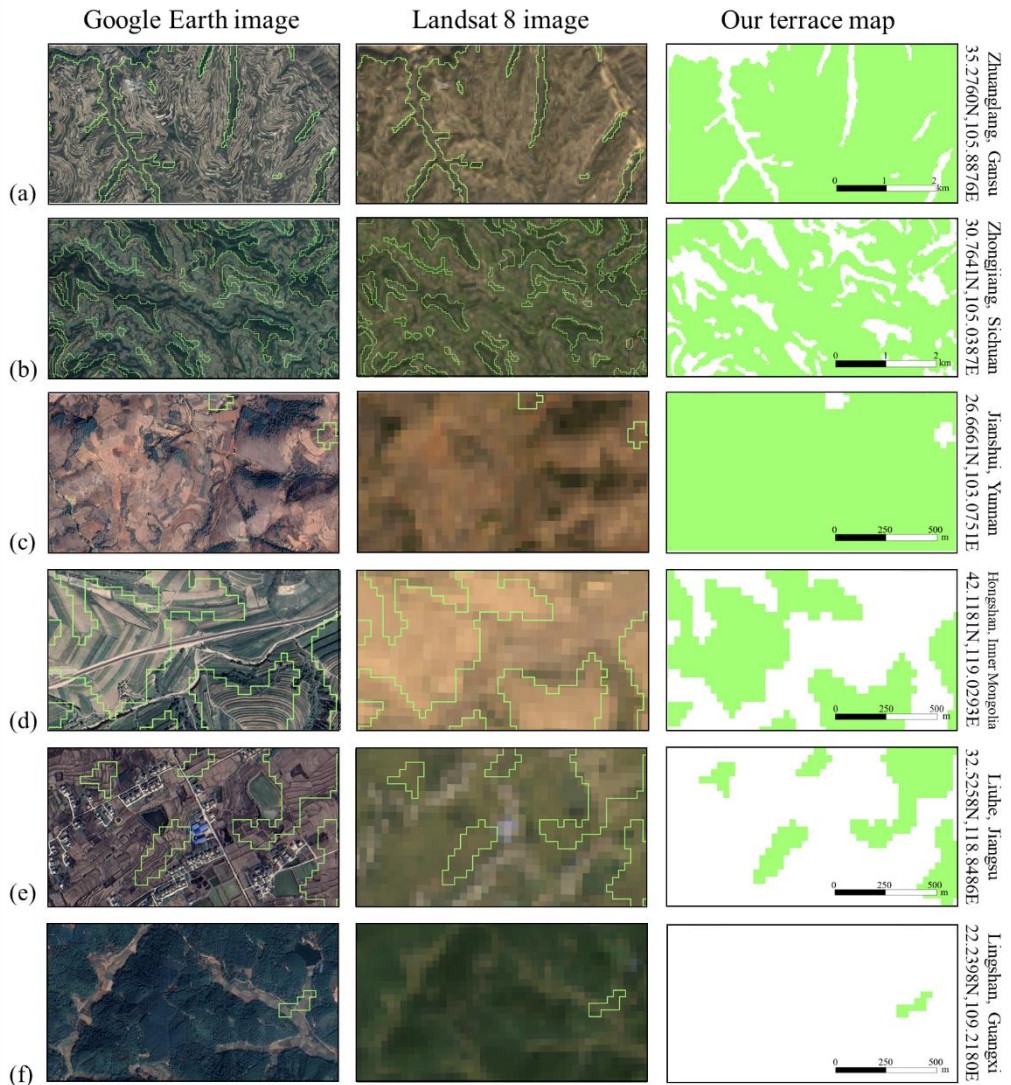


**Figure 11: Selected case comparison of © Google Earth images, Landsat 8 images and our terrace map (green represents terrace and white represents non-terrace) at six different locations: (a)-(f). (a) and (b) are well classified cases, and (c)-(f) are poorly classified cases. Note that they have different scales indicated on the right-hand plots. The green lines in Google Earth images and Landsat 8 images are the corresponding terrace boundaries in our terrace map. The geographic coordinate indicates the image**
**center.**

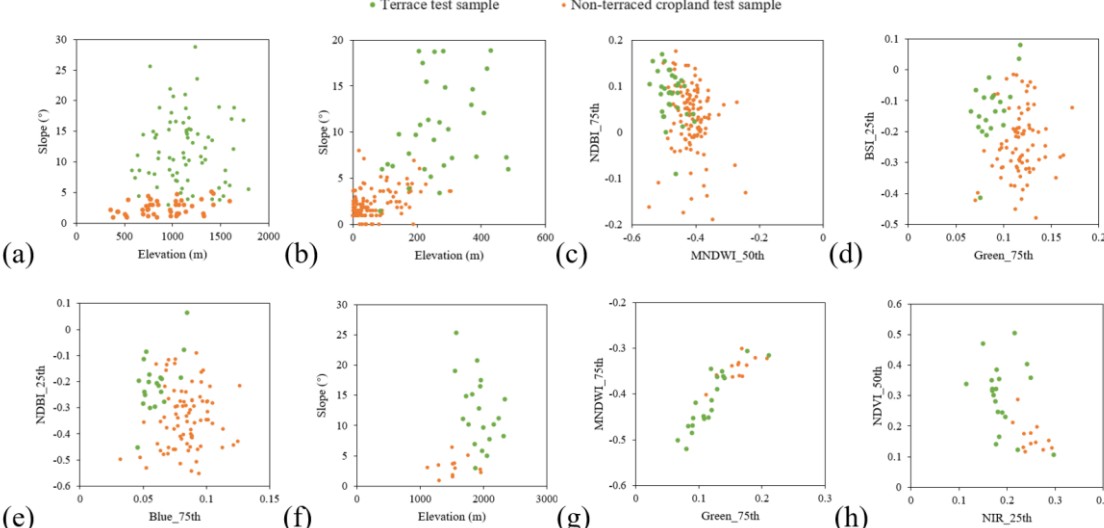

**Figure 12: Scatter plots for distinct features of terrace test samples and non-terraced cropland test samples in four provinces: (a) Shanxi, (b)-(c) Shandong, (d)-(e) Henan, (f)-(h) Ningxia.**


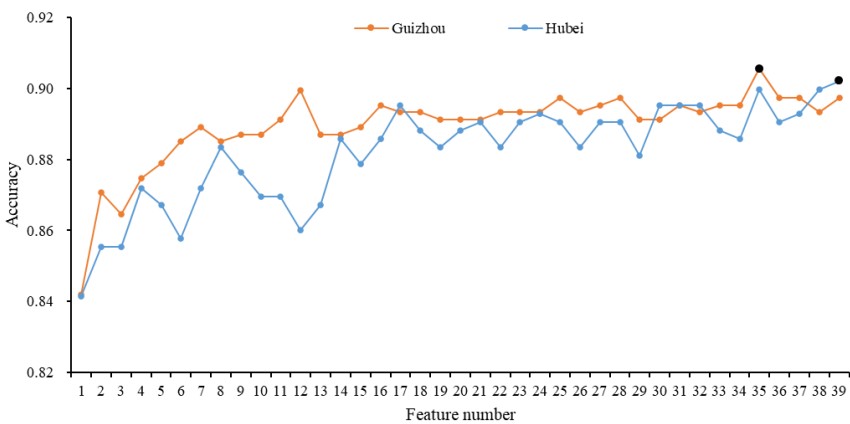

**Figure 13: OA for using different feature numbers in Guizhou and Hubei. The feature addition order along the horizontal axis is identical with the feature importance ranking of the province. The maximum value of accuracy is marked in black.**