# Peer review of "A 30-meter terrace mapping in China using Landsat 8 imagery and digital elevation model based on the Google Earth Engine"

_Earth System Science Data, 2020_

## Short Comment (SC1) · 26 Dec 2020

How do the authors consider this issue of using the elevation layer as a featured band. Since the elevation could have big difference in different province of China, for instance, the average elevation may be 4000m in Gansu province and 300 m in Fujian province (please overlook the number).

---

## Author Comment (AC1) · 8 Jan 2021

Thank you very much for the comments. We agree that the elevation of different provinces varies greatly. To solve the problem, we performed the classification separately for each province in this study. Different training samples were used to train the random forest model of each province and each model was applied to classify the corresponding province. The details can be found in section 2.4.

---

## Referee Comment (RC1) · Anonymous Referee #1 · 22 Jan 2021

General comment:

Terrace extraction is one of the important applications in remote sensing imagery processing in recent decades. Very few research focuses on terrace extraction in large scale areas. The paper suggested a new strategy that collected national, regional and local training samples that could be applied to large-scale areas. However, the effectiveness of the strategy is needed to clarify (Please refer Specific comments, No. 4).

Specific comments:

1. Since the title of the paper is to classify terrace, the accuracy of terrace should be

mentioned in the abstract section.

2. Line 147-148: How did you use the national samples "for training general classification rules and identifying terraces with typical features" in the paper?

3. Line 168-169: What is "the feature number"? Is it the number of training samples?

4. Line 171-172: Is it necessary to collect national and regional training samples instead of only local training samples since you merged the three sample sets and applied each random forest model for each province individually? Please try to apply your model using only the local training samples and compare the results.

5. Table 2, 3; line 214-216: Please clarify your point by describing Statistically significant (P value) of the results of using the two test sample types.

---

## Referee Comment (RC2) · Anonymous Referee #2 · 2 Mar 2021

Review ESSD-2020-157, terraces

Overall, very useful, a good product for ESSD. With some small efforts authors could improve data access and data description for many users.

Section 2.1.1: Authors downloaded a LandSat product that already included cloud masking. But the they applied their own additional cloud masking? Please clarify. Why and how did they perform the second cloud masking.

Section 2.1.2: STRM DEM - provide a citation for the (standard?) void-filling process, here and/or in Table S2. Many readers will not know these acronyms?

Line 112, 113 "we suppose the terrain changes little in decades." Interesting, this reviewer tends to agree. But what decadal changes might impact terraces? Consolidation? Urbanization? One supposes most terraces lie above floodplains, but construction of dams for agricultural water or flood management? In a dynamic Chinese economy, can we really assume static distribution and abundance of terraced agriculture? With their expertise, perhaps authors can and should comment? I note this as possible uncertainty because a few paragraphs later the authors mention use of / calculation of Normalized Difference Building Index (NDBI), which implies that they might need to account for temporal changes in land use over their time periods?

Section 2.1.3, GlobeLand30 represents an interesting product, perhaps unfamiliar to global LULUC communities. Need an actual citation? GlobeLand30 is not in GEE? Same issue about static vs dynamic: in rapidly-changing Chinese economy, can one accept the assumption of no change between 2010 and 2018?

The reviewer confirms 39 features from Table 1: 7 spectral bands from LandSat plus four indices, each with three ranges, plus 6 elevation-related features from SRTM. But several indices listed, e.g. NDBI, incorporate middle infrared (MIR) while MIR not listed among the bands downloaded or processed. Please can authors explain?

Line 153, 154: "(2151 terrace samples and 2639 non-terrace samples) were collected by visual interpretation of Landsat images, SRTM DEM data, cropland extent data extracted from GlobeLand30, and Google Earth images". Authors inspected nearly 5000 images visually? To confirm terrace vs non-terrace? To confirm other land use (agriculture) feature? Impressive but difficult to understand or replicate? What did inspectors gain from GE images not available from LandSat or SRTM? Google Earth images within GEE or separate? GE images one of the least replicatable aspects of this research?

Lines 162, 163: test dataset has a much higher ratio on non-terrace to terrace (almost 10 to 1) than training data (more like 5 to 4). Does this matter? Will this difference arise later as an uncertainty factor?

Section 2.4 Terrace/non-terrace. Because not more than 20% Chinese land area = agriculture, terrace area also has to be << 20%? RF classifier specific to each province - very positive! All this done within GEE?

Section 2.5 Post-processing. Understand the purpose for and mechanism of mode filtering. More interested - and worried - about the sieving process. Terraces on slopes often have a long axis following contour lines but a short axis up- or down-slope. Which axis and/or what areal dimensions do the authors use here as "small". "Small" area for terraces not the same as "small" area of non-terraces? Confused about pixels vs resolution. Understand LandSat at 30

m, but that means 10 pixels represents 300m? Rather large dimension for a "small" area? Or perhaps not on 1km x 1km scale? Please correct my mis-impressions.

Line 204: Because, in subsequent text, authors often switch between user/producer terms (UA, PA) and commission/omission terms, take this opportunity to define both? For example, in line 204, "the user's accuracy (UA) of the terrace class" could become 'the user's accuracy (UA, also referred to as 'commission error') of the terrace class'. Authors will know how best to make these connections but because they use both (user/producer and commission/omission) they need to clarify.

Line 211 - again, reader needs to know explicitly how authors define "small".

Line 212 - is this second test set of 301 samples also available at Zenodo? Not obvious? It seems like the authors regard it as an important independent test product?

Line 240 - if cropland represents 20% of total China land area, and terraces represent 26 to 28% of cropland, then terraces represent 5% to almost 6% of total China area? I do this as a sort of 'mass balance' check; have I got this correct? If not correct, what did I miss?

Line 277 - finally a list of 12 excluded provinces! Having this information appear earlier in the manuscript would have answered many questions for this reader.

Uncertainty assessment/discussion and feature importance, including Figures 6 to 10, represent strong positive features of this data description. Thank authors for this careful analysis.

Limitations and directions - overall a very useful discussion, particularly about spatial resolution. Again this issue of visual inspection. Who has energy to visually inspect 1000s of images and how does one quality-control outcomes of such inspection? Even if the authors can not provide quantitative assessment, they could help other users with a general estimate of the effort involved? The authors also discuss merits of GlobeLand30 vs other LU products. Again, users will want to know how they might get access to GL30 (e.g. via GEE?) as well as other references to GL30 use and accuracy.

On Fig 3, not obvious that Xinjiang or Heilongjiang provinces have terraces. Need different or brighter color scale. Could authors mark location of positive outcomes shown in Figure 4 in Yunnan, Hunan or Guangxi provinces on Figure 3?

Figure 5 does not include all the provinces show in map on Figure 3? By cropland or terrace abundance or some other LU factor, authors have eliminated far west or far north provinces from their analysis. Did the reviewer miss an earlier statement to this effect? These represent the "other provinces" mentioned in line 238? No, they represent the 12 province excluded, listed at line 277. A reader needs to see this exclusion information earlier, before any results, perhaps even before most methods?

Figure 10: very difficult to read / interpret axis labels. Table S1 helps, it should move into main manuscript as part of legend for Figure 10?

(Because Table S2, the source attribution table, also belongs in main manuscript, perhaps in or near Section 2, authors could include S1 and S2 in main manuscript and thereby eliminate supplement?)

---

## Author Comment (AC2) · 15 Mar 2021

**Response to comments**

**Title**: A 30-meter terrace mapping in China using Landsat 8 imagery and digital elevation model based on the Google Earth Engine

**MS No.**: essd-2020-157

5

**Referee #1**

**General comments**

**Comment 1:**

Terrace extraction is one of the important applications in remote sensing imagery processing in recent decades. Very few
10 research focuses on terrace extraction in large scale areas. The paper suggested a new strategy that collected national, regional and local training samples that could be applied to large-scale areas. However, the effectiveness of the strategy is needed to clarify (Please refer Specific comments, No. 4).

**Response 1:**

Thank you very much for the comments and suggestions. Please see the detailed point-by-point responses below.

15

**Specific comments**

**Comment 1:**

Since the title of the paper is to classify terrace, the accuracy of terrace should be mentioned in the abstract section.

**Response 1:**

20 Thank you for your advice. We added the accuracy of terrace class in **abstract**: "For terrace class, the producer's accuracy (PA) was 79.945% and the user's accuracy (UA) was 71.149%."

**Comment 2:**

Line 147-148: How did you use the national samples "for training general classification rules and identifying terraces with
25 typical features" in the paper?

**Response 2:**

The random forest classifier can learn the terrace/non-terrace characteristics and summarize the classification rules based on the features of samples. So, the characteristics and rules RF learned were similar to those of the samples. Because the national terrace samples were collected from the terraces with typical features, the trained classifier can identify the typical terraces.

30 The specific principle of training and classification using random forest classier was supplemented in **Section 2.4**: "It is consisted of multiple decision trees, all of which perform classification separately and vote for the final results. During the training process of decision tree, each tree node is split based on the most contributing feature among the randomly selected input features of the training sample subset. After training, each decision tree judges the pixel class according to the established tree rules (Breiman, 2001; Gislason et al., 2006)."

35

**Comment 3:**

Line 168-169: What is "the feature number"? Is it the number of training samples?

**Response 3:**

In machine learning, feature is the selected property or characteristic of samples. The feature number is 39 in this study.
40 Detailed information about features used in the study can be found in Section 2.2.

To make it clearer, we added the specific number of features: "the number of decision trees and the number of variables per split, were set to 200 and the rounded square root of the feature number (39)" (**Section 2.4**).

**Comment 4:**

45 Line 171-172: Is it necessary to collect national and regional training samples instead of only local training samples since you merged the three sample sets and applied each random forest model for each province individually? Please try to apply your model using only the local training samples and compare the results.

**Response 4:**

Thank you for the suggestion. In the study, we adopted the strategy of collecting national, regional and local samples in order
50 to improve the sampling efficiency, i.e., reduce the workload of sampling through reusing the national and regional samples. As described in Section 2.3, we first collected the national and regional samples (801 samples and 54 samples respectively), and then supplemented some local samples (3989 samples) to improve the mapping accuracy of all provinces except Macao, where there is no cropland. However, the local sample size was very small (N<10 for either terrace or non-terrace) in some provinces (Gansu, Guizhou, Heilongjiang, Hongkong, Jilin, Macau, Ningxia, Shandong, Shanghai, Tibet, Xinjiang, Yunnan).
55 So, only using the local samples collected in the study is insufficient for training the classifier and comparing the results.

We supplemented some sentences in **Section 2.3** to clarify the purposes of collecting national, regional and local samples: "Through reusing the national and regional samples, smaller total sample size was required and the workload of sampling was minimized effectively.".

**Comment 5:**

Table 2, 3; line 214-216: Please clarify your point by describing Statistically significant (P value) of the results of using the two test sample types.

**Response 5:**

We calculated the P-value as the reviewer suggested to prove the accuracies of terrace class evaluated by the two test sample sets are similar.

We added the related texts to describe statistically significant in **Section 3.2**: "The accuracy evaluation result using the 301 test samples of known terraces (Table 4) was numerically similar to the above result using the 10875 random test samples (Table 3). The Chi-square tests (Mantel, 1963) were carried out for the two PAs and UAs of terrace class respectively to further prove the similarity quantitatively. The P-values of both tests were greater than 0.05, indicating there was no statistically significant difference between the terrace accuracies using the two test sample sets.".

**Reference:**

*Mantel, N.: Chi-square tests with one degree of freedom; extensions of the Mantel-Haenszel procedure, J. Am. Stat. Assoc., 58, 690-700, https://doi.org/10.1080/01621459.1963.10500879, 1963.*

---

## Author Comment (AC3) · 15 Mar 2021

**Response to comments**

**Title**: A 30-meter terrace mapping in China using Landsat 8 imagery and digital elevation model based on the Google Earth Engine

**MS No.**: essd-2020-157

**Referee #2**

**General comments**

**Comment 1:**

Overall, very useful, a good product for ESSD. With some small efforts authors could improve data access and data description

10   for many users.

**Response 1:**

Thank you very much for the comments and suggestions. Please see the detailed point-by-point responses below.

**Specific comments**

15   **Comment 1:**

Section 2.1.1: Authors downloaded a LandSat product that already included cloud masking. But they applied their own additional cloud masking? Please clarify. Why and how did they perform the second cloud masking.

**Response 1:**

Sorry for the unclear expression. The Landsat product in the Google Earth Engine just includes the cloud mask band (produced

20   using CFMASK) to indicate the cloudy pixels, but the cloud masking is not performed. We conducted the cloud masking according to the cloud mask band (the pixel_qa band).

We modified the original sentence to "This product has been atmospherically corrected and contains a cloud, shadow, water, and snow mask band produced using CFMASK, as well as a per-pixel saturation mask band (USGS, 2018). All scenes covering China acquired in 2018 were selected in our study. After obtaining the 10196 images, we removed the clouds in each image

25   based on the cloud mask band (the pixel QA band) of Landsat 8 SR data." (**Section 2.1.1**).

**Comment 2:**

30 Section 2.1.2: STRM DEM - provide a citation for the (standard?) void-filling process, here and/or in Table S2. Many readers will not know these acronyms?

**Response 2:**

We supplemented a reference for the void-filling process in **Section 2.1.2**: "The product at 1 arc-second (30 m) resolution is available on GEE, which has undergone a void-filling process using open-source data (ASTER GDEM2, GMTED2010, and
35 NED) (USGS, 2015).".

**Reference:**

*USGS.: The shuttle radar topography mission (SRTM) collection user guide, https://lpdaac.usgs.gov/documents/179/SRTM_User_Guide_V3.pdf, last access: 15 March 2021, 2015.*
* * *
40 ## Comment 3:

Line 112, 113 "we suppose the terrain changes little in decades." Interesting, this reviewer tends to agree. But what decadal changes might impact terraces? Consolidation? Urbanization? One supposes most terraces lie above floodplains, but construction of dams for agricultural water or flood management? In a dynamic Chinese economy, can we really assume static distribution and abundance of terraced agriculture? With their expertise, perhaps authors can and should comment? I note this
45 as possible uncertainty because a few paragraphs later the authors mention use of / calculation of Normalized Difference Building Index (NDBI), which implies that they might need to account for temporal changes in land use over their time periods?

**Response 3:**

We do not and can not assume the static distribution and abundance of terraced agriculture. As a land use/cover type, terrace may transform with other land use/cover types (e.g., slope cropland, urban, other vegetations). As you mentioned, both
50 construction of dams or other facilities and urbanization can change terrace distribution. However, most land use/cover changes will not have much impact on the terrain.

We supplemented the statement about the uncertainty caused by terrain data in **Section 3.6**: "On the one hand, the inconsistent year of terrain data and classification led to the uncertainties. Terrain may change due to the land use/cover transformations during 2010-2018. However, in relative to the vertical accuracy of terrain data, most transformations had little impact on terrain.
55 Even if terrain changed significantly in somewhere during the eight years, the spectral features in 2018 can help with classification. And the satisfactory accuracy of our terrace map also indicated the assumption of little terrain change was acceptable. But there is no doubt that better results can be achieved if high resolution and precision terrain data in 2018 is available."
* * *
60

**Comment 4:**

Section 2.1.3, GlobeLand30 represents an interesting product, perhaps unfamiliar to global LULUC communities. Need an actual citation? GlobeLand30 is not in GEE? Same issue about static vs dynamic: in rapidly-changing Chinese economy, can one accept the assumption of no change between 2010 and 2018?

**Response 4:**

GlobeLand30 is not in GEE, we downloaded the product from *www.globeland30.org* and uploaded it into GEE.

Both reference and information related to GlobeLand30 were added to the manuscript: "a well-established and widely used source of land cover information, generated by integration of pixel-based and objected-based methods with knowledge (POK) using multi-source data (Chen et al., 2015)." (**Section 2.1.3**); "GlobeLand30 was first downloaded from the website (www.globeland30.org) and then uploaded into GEE." (**Section 2.1.3**).

**Reference:**

*Chen, J., Chen, J., Liao, A., Cao, X., Chen, L., Chen, X., He, C., Han, G., Peng, S., Lu, M., Zhang, W., Tong, X. and Mills, J.: Global land cover mapping at 30 m resolution: A POK-based operational approach, ISPRS J. Photogramm. Remote Sens., 103, 7-27, https://doi.org/10.1016/j.isprsjprs.2014.09.002, 2015.*

Cropland was indeed changing between 2010 and 2018, but we found it had little influence to our results. We added the quantification of the uncertainty caused by the inconsistent years to **Section 3.6**: "Furthermore, the non-correspondence of cropland extent data year and terrace/non-terrace classification year also had an impact on the results. Compared to using a cropland map in 2018, the major limitation of using cropland map in 2010 is that some terraces located in the newly increased cropland area will be omitted. We quantified the omission caused by the cropland mask using the test samples. Only 119 of the total 1092 terrace test samples located outside the cropland extent of GlobeLand30 2010, indicating the maximum possible omission errors caused by the non-corresponding year was 10.90%.".

**Comment 5:**

The reviewer confirms 39 features from Table 1: 7 spectral bands from LandSat plus four indices, each with three ranges, plus 6 elevation-related features from SRTM. But several indices listed, e.g. NDBI, incorporate middle infrared (MIR) while MIR not listed among the bands downloaded or processed. Please can authors explain?

**Response 5:**

Sorry for mistakes. In the table, the MIR band in the algorithm of NDBI and MNDWI corresponds to the SWIR1 band of Landsat 8 surface reflectance (SR) imagery.

We have revised the algorithms in **Table 2**.

 **Table 2: Features for terrace/non-terrace classification.**

| Feature | Data source/Algorithm |
|---|---|
| *Percentiles of spectral bands/indices* | *Landsat 8 surface reflectance (SR) imagery* |
| Bands | Landsat 8 SR Band 2—7 (Blue, Green, Red, NIR, SWIR1, SWIR2), Band 11 (TIRS2) |
| Normalized Difference Vegetation Index (NDVI) | $\frac{(NIR - Red)}{(NIR + Red)}$ |
| Modified normalized Difference Water Index (MNDWI) | $\frac{(Green - SWIR1)}{(Green + SWIR1)}$ |
| Normalized Difference Building Index (NDBI) | $\frac{(SWIR1 - NIR)}{(SWIR1 + NIR)}$ |
| Bold Soil Index (BSI) | $\frac{((SWIR1 + Red) - (Blue + NIR))}{((SWIR1 + Red) + (Blue + NIR))}$ |
| *Topographic factors* | *Shuttle Radar Topography Mission digital elevation model (SRTM DEM) data* |
| Elevation | SRTM DEM data |
| Slope | $\frac{Elevation\ change}{Horizontal\ distance\ change}$ |
| Slope of Slope (SOS) | $\frac{Slope\ change}{Horizontal\ distance\ change}$ |
| Roughness (R) | $\frac{S_{curved\ surface}}{S_{plane\ surface}}$ |
| Slope shape (P) | $H_{i,j} - \frac{\sum_{i=1}^{n} H_i}{n}$ |
| Relief (RF) | $H_{max} - H_{min}$ |

**Comment 6:**

Line 153, 154: "(2151 terrace samples and 2639 non-terrace samples) were collected by visual interpretation of Landsat images, SRTM DEM data, cropland extent data extracted from GlobeLand30, and Google Earth images". Authors inspected nearly 5000 images visually? To confirm terrace vs non-terrace? To confirm other land use (agriculture) feature? Impressive but difficult to understand or replicate? What did inspectors gain from GE images not available from LandSat or SRTM? Google Earth images within GEE or separate? GE images one of the least replicatable aspects of this research?

**Response 6:**

Yes, we visually interpreted the images to confirm terraces and non-terraces according to land use features. The process of sample collection is random and difficult to replicate. From high-resolution GE images, we can gain more detailed information, such as texture. GE images are especially useful for us to visually identify small terraces. In the study, we used the Google

Earth images within GEE. Compared with other images such as Landsat images and SRTM images, GE images are less replicable, but some history GE images can be viewed in the Google Earth software.

110   We added **Section 2.1.4** to describe Google Earth images used in the study: "Google Earth images on GEE were used as auxiliary data for samples collection. This dataset is a composited product combining multiple sets of satellite imagery, which are provided by different commercial image providers or government agencies at different zoom level (Potere, 2008). Its highest resolution can reach less than 1 m. With more detailed information (e.g., texture) provided by the high-resolution Google Earth images, we can visually distinguish the samples more accurately.".

115   **References:**

*Potere, D.: Horizontal positional accuracy of Google Earth's high-resolution imagery archive, Sensors, 8, 7973-7981, https://doi.org/10.3390/s8127973, 2008.*

**Comment 7:**

120   Lines 162, 163: test dataset has a much higher ratio on non-terrace to terrace (almost 10 to 1) than training data (more like 5 to 4). Does this matter? Will this difference arise later as an uncertaintly factor?

**Response 7:**

It is true that using sample sets with different sample ratios can get different accuracy evaluation results. Generally, the accuracy will be more reliable when the sample size ratio of terrace/non-terrace is closer to the real area ratio of terrace/non-

125   terrace and the sample sizes of both types are sufficient. Our test samples were randomly generated in order to approach the real area ratio. And the subsequent two rounds of sample densifying described in Section 2.3 further ensured enough terrace samples (N>1000). Thus, we think the test sample size ratio is appropriate and can reflect the true accuracy of mapping result.

**Comment 8:**

130   Section 2.4 Terrace/non-terrace. Because not more than 20% Chinese land area = agriculture, terrace area also has to be << 20%? RF classifier specific to each province - very positive! All this done within GEE?

**Response 8:**

Right. The area information was supplemented to **Section 3.3**: "As for the total area of terraces in China for 2018, it was estimated to be 53.55 Mha by the PC method, accounting for 26.43% of China's cropland area and 5.58% of China's land area

135   (about 960 Mha). And the EM method showed that the total terrace area was 58.46±2.99 Mha, i.e., 28.85%±1.48% of China's cropland area and 6.09%±0.31% of China's land area.".

In addition to the whole mapping process, the area calculation was also done within GEE.

140

**Comment 9:**

Section 2.5 Post-processing. Understand the purpose for and mechanism of mode filtering. More interested - and worried - about the sieving process. Terraces on slopes often have a long axis following contour lines but a short axis up- or down-slope. Which axis and/or what areal dimensions do the authors use here as "small". "Small" area for terraces not the same as "small" area of non-terraces? Confused about pixels vs resolution. Understand LandSat at 30 m, but that means 10 pixels represents 300m? Rather large dimension for a "small" area? Or perhaps not on 1km x 1km scale? Please correct my mis-impressions.

**Response 9:**

In the study, patch area was used to measure "small". Terrace/non-terrace patch with an area of 10 pixels (i.e., about 9000 m$^2$) or less was considered as "small". The threshold (10 pixels) was determined through experiments. We tried a series of sieving threshold and got the highest accuracy when the threshold was set to 10 pixels.

Several sentences were added to make it more clear in **Section 2.5.2**: "Namely, terrace/non-terrace patches with an area of 10 pixels (about 9000 m$^2$) or less were sieved.".

**Comment 10:**

Line 204: Because, in subsequent text, authors often switch between user/producer terms (UA, PA) and commission/omission terms, take this opportunity to define both? For example, in line 204, "the user's accuracy (UA) of the terrace class" could become 'the user's accuracy (UA, also referred to as 'commission error') of the terrace class'. Authors will know how best to make these connections but because they use both (user/producer and commission/omission) they need to clarify.

**Response 10:**

Thank you for the advice. We also agree the connections will make it easier for readers to understand.

We added the connections in **Section 3.2**: "The producer's accuracy (PA, also referred to as "1-omission error") of the terrace class was 79.945%, whereas the user's accuracy (UA, also referred to as "1-commission error") of the terrace class was 71.149%".

**Comment 11:**

Line 211 - again, reader needs to know explicitly how authors define "small".

**Response 11:**

Here, small was also defined as "terraces with an area of 10 pixels (i.e., about 9000 m$^2$) or less", which were sieved during the post-processing, and thus could not be identified in our terrace map.

In **Section 3.2**, the explicit definition "small patch terraces (terrace with an area of 10 pixels (about 9000 m$^2$) or less)" was given.

**Comment 12:**

Line 212 - is this second test set of 301 samples also available at Zenodo? Not obvious? It seems like the authors regard it as an important independent test product?

**Response 12:**

Sorry, the test sample set is not available at Zenodo. The sample sets are important for mapping project. Please understand that we can not make them public because of some cooperation.

**Comment 13:**

Line 240 - if cropland represents 20% of total China land area, and terraces represent 26 to 28% of cropland, then terraces represent 5% to almost 6% of total China area? I do this as a sort of 'mass balance' check; have I got this correct? If not correct, what did I miss?

**Response 13:**

Correct. The area information was supplemented to **Section 3.3**: "As for the total area of terraces in China for 2018, it was estimated to be 53.55 Mha by the PC method, accounting for 26.43% of China's cropland area and 5.58% of China's land area. And the EM method showed that the total terrace area was 58.46±2.99 Mha, i.e., 28.85%±1.48% of China's cropland area and 6.09%±0.31% of China's land area."

**Comment 14:**

Line 277 - finally a list of 12 excluded provinces! Having this information appear earlier in the manuscript would have answered many questions for this reader.

**Response 14:**

Thank you for the advice. We added the information about test samples in the previous part of the manuscript (**Section 2.3**): "The terrace test sample is zero in 12 provinces (Beijing, Hainan, Heilongjiang, Hongkong, Jilin, Jiangsu, Macao, Shanghai, Taiwan, Tianjin, Tibet, and Xinjiang), while the terrace/non-terrace test samples are insufficient (N<10 for either terrace or non-terrace) in 14 provinces (Liaoning, Zhejiang, and the above 12 provinces). Thus, terrace area of the 14 provinces was not analyzed in Section 3.3 and accuracy of the 12 provinces was not evaluated in Section 3.4.3.".

**Comment 15:**

Uncertainty assessment/discussion and feature importance, including Figures 6 to 10, represent strong positive features of this data description. Thank authors for this careful analysis.

**Response 15:**

205 Thank you for the acknowledgement and encouragement.

**Comment 16:**

Limitations and directions - overall a very useful discussion, particularly about spatial resolution. Again this issue of visual inspection. Who has energy to visually inspect 1000s of images and how does one quality-control outcomes of such inspection?
210 Even if the authors can not provide quantitative assessment, they could help other users with a general estimate of the effort involved? The authors also discuss merits of GlobeLand30 vs other LU products. Again, users will want to know how they might get access to GL30 (e.g. via GEE?) as well as other references to GL30 use and accuracy.

**Response 16:**

The sample collection work referred to a previous study (Zhao et al., 2014). We double-checked the samples to ensure a
215 quality-control outcome. Information related to sample collection was added in **Section 2.3**: "We referred to Zhao et al. (2014) to conduct the interpretation and quality control. The samples were double-checked to ensure reliability.".

The GlobeLand30 product can be accessed in *www.globeland30.org*. We downloaded it from the website and uploaded it to the GEE. More detailed information about the GlobeLand30 can be accessed from the references.

Information related to GlobeLand30 was added to the manuscript: "a well-established and widely used source of land cover
220 information, generated by integration of pixel-based and objected-based methods with knowledge (POK) using multi-source data (Chen et al., 2015)." (**Section 2.1.3**); "GlobeLand30 was first downloaded from the website (www.globeland30.org) and then uploaded into GEE." (**Section 2.1.3**);

**Reference:**

*Zhao, Y., Gong, P., Yu, L., Hu, L., Li, X., Li, C., Zhang, H., Zheng, Y., Wang, J., Zhao, Y., Cheng, Q., Liu, C., Liu, S. and Wang,*
225 *X.: Towards a common validation sample set for global land-cover mapping. Int. J. Remote Sens., 35, 4795-4814, https://doi.org/10.1080/01431161.2014.930202, 2014.*

*Chen, J., Chen, J., Liao, A., Cao, X., Chen, L., Chen, X., He, C., Han, G., Peng, S., Lu, M., Zhang, W., Tong, X. and Mills, J.: Global land cover mapping at 30 m resolution: A POK-based operational approach, ISPRS J. Photogramm. Remote Sens., 103, 7-27, https://doi.org/10.1016/j.isprsjprs.2014.09.002, 2015.*

230

**Comment 17:**

On Fig 3, not obvious that Xinjiang or Heilongjiang provinces have terraces. Need different or brighter color scale. Could authors mark location of positive outcomes shown in Figure 4 in Yunnan, Hunan or Guangxi provinces on Figure 3?

**Response 17:**

235 In our terrace map, there are very few terraces in Xinjiang and Heilongjiang provinces. Thus, it is difficult to find them in Fig 3. To make it easier for readers to find them in Fig 3, we changed the display resolution of terrace map in Xinjiang, Heilongjiang and Liaoning provinces from 1km to 5km. And we also marked the terrace locations in Figure 4 on Figure 3.

The revised **Figure 3** is shown below.

[Figure]

**Figure 3: Terrace distribution across China in 2018. The map values indicate the proportion of terrace within a 1km×1 km grid cell except for Heilongjiang, Liaoning and Xinjiang, where the mapping results are displayed at 5×5 km for clearer visual effect. Shanghai and Macao are the only two provinces have no terrace in this map. The locations of three well-known terraces shown in Fig. 4 are marked as stars in the terrace map.**

**Comment 18:**

Figure 5 does not include all the provinces show in map on Figure 3? By cropland or terrace abundance or some other LU factor, authors have eliminated far west or far north provinces from their analysis. Did the reviewer miss an earlier statement to this effect? These represent the "other provinces" mentioned in line 238? No, they represent the 12 province excluded, listed at line 277. A reader needs to see this exclusion information earlier, before any results, perhaps even before most methods?

**Response 18:**

Yes, Figure 3 includes all the provinces in China (34 provinces), but Figure 5 does not include the 14 provinces without sufficient test samples (N<10 for either terrace or non-terrace).

We explained the "other provinces" in **Section 3.3**: "Terrace area for other 14 provinces (Beijing, Hainan, Heilongjiang, Hongkong, Jilin, Jiangsu, Liaoning, Macao, Shanghai, Taiwan, Tianjin, Tibet, Xinjiang, and Zhejiang) have not been analyzed due to insufficient test samples to estimate the uncertainty".

We also added the exclusion information in the previous part (**Section 2.3**): "The terrace test sample is zero in 12 provinces (Beijing, Hainan, Heilongjiang, Hongkong, Jilin, Jiangsu, Macao, Shanghai, Taiwan, Tianjin, Tibet, and Xinjiang), while the terrace/non-terrace test samples are insufficient (N<10 for either terrace or non-terrace) in 14 provinces (Liaoning, Zhejiang,

and the above 12 provinces). Thus, terrace area of the 14 provinces was not analyzed in Section 3.3 and accuracy of the 12
260    provinces was not evaluated in Section 3.4.3.".
* * *
**Comment 19:**

Figure 10: very difficult to read / interpret axis labels. Table S1 helps, it should move into main manuscript as part of legend

for Figure 10?

265    (Because Table S2, the source attribution table, also belongs in main manuscript, perhaps in or near Section 2, authors could

include S1 and S2 in main manuscript and thereby eliminate supplement?)

**Response 19:**

Thank you for the suggestion. We changed Table S1 to Table 5 and moved it next to Figure 10 in **Section 3.5**. And we changed

Table S2 to Table 1 in **Section 2.1**. Accordingly, the sequence numbers of other tables were changed in the manuscript.

270

---

## Referee Report (RR1)

**Review the revised version**

**Title:** A 30-meter terrace mapping in China using Landsat 8 imagery and digital elevation model based on the Google Earth Engine

Thank you for your detailed responses.

Below are my comments for the revised version of your paper, please refer to them.

**1. Response 3:** In machine learning, feature is the selected property or characteristic of samples. The feature number is 39 in this study. Detailed information about features used in the study can be found in Section 2.2.

**Comment:** Were all 39 input features useful for the rice terrace classification? It was time-consuming to prepare the input feature as well as train the model using the huge input data, wasn't it?

**2. Response 4:** …However, the local sample size was very small (N<10 for either terrace or non-terrace) in some provinces (Gansu, Guizhou, Heilongjiang, Hongkong, Jilin, Macau, Ningxia, Shandong, Shanghai, Tibet, Xinjiang, Yunnan). So, only using the local samples collected in the study is insufficient for training the classifier and comparing the results.

**Comment:** What if you apply the model using only local samples and compare the results for the provinces with sufficient train and test samples?

**3. "Nine-Dash Line":** In Figures 2, 3, 6, 9, other I will have missed, the controversial "Nine-Dash Line" is shown. I strongly recommend the authors to remove the controversial "Nine-Dash Line" from all figures of this manuscript. This item is irrelevant to the scientific content of this paper, and has also been rejected by a 2016 international tribunal in The Hague (see a summary at https://www.theguardian.com/news/2016/jul/12/south-china-sea-dispute-what-you-need-to-know-about-the-hague-court-ruling?fbclid=IwAR0POoX2gUpHd_r16bFtpEUKwkxaY23z4du1Dbqq0IqpEV6IDQ7HJh6k8jk; and the Press Release of this international court at https://pcacases.com/web/sendAttach/1801).

I strongly believe that papers published in ESSD journal should only focus on the scientific aspects of the Land Cover and Land Use mapping disciplines rather than (political) propaganda. As a result, the inclusion of the "Nine-dash line" is both irrelevant and inappropriate.

---

## Author Response (AR2)

**Response to comments**

**Title**: A 30-meter terrace mapping in China using Landsat 8 imagery and digital elevation model based on the Google Earth Engine

**MS No.**: essd-2020-157

5

**Referee #1**

**General comments**

**Comment 1:**

Below are my comments for the revised version of your paper, please refer to them.

10 **Response 1:**

Thank you very much for the comments and suggestions. Please see the detailed point-by-point responses below.

**Specific comments**

**Comment 1:**

15 Were all 39 input features useful for the rice terrace classification? It was time consuming to prepare the input feature as well as train the model using the huge input data, wasn't it?

**Response 1:**

Thank you for your comments. As described in Section 2.2, all these features play an important role in terrace/non-terrace classification. And due to the difficulty of terrace identification and large area of research region, it is necessary to use more 20 features. Additionally, we also further quantified the impact of feature number on classification accuracy and added the statement in **Section 3.6**: "To further illustrate the usefulness of all the 39 features selected in the study, we took a terraced cropland-dominated province (Guizhou) and a non-terraced cropland-dominated province (Hubei) as examples to train the classification model based on different feature numbers and evaluate the accuracy. According to Fig. 13, OA generally showed an upward and gradually stable trend as the feature number increased in both provinces, the maximum values were reached 25 when using 35 features in Guizhou and 39 features in Hubei, indicating features were not redundant. Therefore, we applied all features in this study."

[Figure]

**Figure 13: OA for using different feature numbers in Guizhou and Hubei. The feature addition order along the horizontal axis is identical with the feature importance ranking of the province. The maximum value of accuracy is marked in black.**

30

As for the efficiency, if using traditional software packages, it is indeed time consuming for data preprocessing and model training. However, the GEE platform provides the analysis-ready data and high-performance parallel computation service, making it convenient and fast to prepare all the 39 features and train the classification model.

35

**Comment 2:**

What if you apply the model using only local samples and compare the results for the provinces with sufficient train and test samples?

**Response 2:**

40 Thank you for your suggestion. We trained the classification model using only the local training samples in all provinces where both local terrace and local non-terrace training sample number were more than 10. And the results were compared with the classification using national, regional and local training samples in the study.

The comparation was supplemented in **Section 3.6**: "As for the sampling strategy, to further clarify the effectiveness of the national and regional training samples in the study, we compared the accuracies of classification through only using local

45 training samples and through using national, regional and local training samples in the provinces where both local terrace and local non-terrace training sample size were more than 10 (Anhui, Beijing, Chongqing, Fujian, Guangdong, Guangxi, Hainan, Hebei, Henan, Hubei, Hunan, Inner Mongolia, Jiangsu, Jiangxi, Liaoning, Qinghai, Shaanxi, Shanxi, Sichuan, Taiwan, Tianjin, Zhejiang). On the whole, adding the national and regional samples increased OA by 6.90% in these provinces, proving our sampling strategy in the study is reliable and can be applied to other large-scale researches."

50

**Comment 3:**

"Nine-Dash Line": In Figures 2, 3, 6, 9, other I will have missed, the controversial "Nine Dash Line" is shown. I strongly recommend the authors to remove the controversial "Nine-Dash Line" from all figures of this manuscript. This item is irrelevant to the scientific content of this paper, and has also been rejected by a 2016 international tribunal in The Hague (see a summary at https://www.theguardian.com/news/2016/jul/12/south-china-sea-dispute-what-you-need-to-know-about-the-hague-court-ruling?fbclid=IwAR0POoX2gUpHd_r16bFtpEUKwkxaY23z4du1Dbqq0IqpEV6IDQ7HJh6k8jk; and the Press Release of this international court at https://pcacases.com/web/sendAttach/1801).

I strongly believe that papers published in ESSD journal should only focus on the scientific aspects of the Land Cover and Land Use mapping disciplines rather than (political) propaganda. As a result, the inclusion of the "Nine-dash line" is both irrelevant and inappropriate.

**Response 3:**

Thank you for your advice. We have removed the "Nine-Dash Line" in all these figures.